# Nuclear deformability increases PARPi sensitivity in BRCA1-deficient cells by increasing microtubule-dependent DNA break mobility

Elena Faustini[1], Angela dello Stritto[1], Andrea Panza[1], Ylli Doksani [2] & Francisca Lottersberger [1] ✉

Microtubules and nuclear transmembrane SUN1/2 proteins promote the mobility of DNA Double Strand Breaks (DSBs) induced by ionizing radiation and the misrepair of one-ended DSBs induced in BRCA1-deficient cells by Poly(ADP-ribose) polymerase inhibitors (PARPi). However, whether microtubules promote aberrant DSBs repair by altering the nuclear structure and whether the nuclear structure itself plays a role in these processes is still unclear. Here we show that microtubule-dependent DSBs mobility in BRCA1-deficient cells after PARPi treatment is associated with nuclear envelope (NE) invaginations. Furthermore, increasing NE invaginations by *Lmna* deletion or inhibition of sphingolipid synthesis increases DSBs mobility, chromosomal aberrations, and PARPi cytotoxicity in BRCA1-deficient cells. These findings reveal a functional connection between the NE and DSB repair and suggest that drugs increasing NE deformability will enhance PARPi therapy efficacy in BRCA1-deficient cancers.

Small-molecule inhibition of Poly (ADP-ribose) polymerase 1 and 2 (PARP1/2) is an approved therapeutic strategy for the treatment of metastatic breast, ovarian, pancreatic and prostate cancers associated with Homologous Recombination (HR) defects, including mutations in *BRCA1* gene. The lethality of PARP inhibitors (PARPi) is explained by the accumulation of chromosomal aberrations due to the mis-repair of one-ended Double Strand Breaks (DSBs) generated by the encounter of the replication machinery with PARP1/2-trapped single-stranded breaks (SSBs)[1,2]. While normally these DSBs can be accurately repaired by HR using the sister chromatid as template, when HR is defective, they became substrate of Non-Homologous End Joining (NHEJ), which ligates one-ended breaks regardless of their chromosomal location. In details, in the absence of BRCA1, the 5′-end resection step, necessary to initiate HR, is prevented by the 53BP1/RIF1/Shieldin/CST-Polα/primase axis[3–19]. Ablation of any of these factors restores resection and,

therefore, HR in the absence of BRCA1, thus promoting survival and increasing resistance to PARP inhibition.

Independent from 5′-end resection, dynamic cytosolic microtubules and the linker of nucleoskeleton and cytoskeleton (LINC) complex Inner Nuclear Membrane (INM) components SUN1 and SUN2 contribute to the accumulation of chromosome rearrangements in PARPi-treated BRCA1-depleted Mouse Embryonic Fibroblasts (MEFs)[20]. Since drugs affecting microtubule dynamics and *Sun1/Sun2* double deletion are epistatic in reducing the mobility of both dysfunctional telomeres and γ-irradiation-induced two-ended DSBs, we previously proposed that microtubules and the LINC complex increase the likelihood of the aberrant joining of PARPi-induced one-ended DSBs by increasing their mobility[20]. In agreement with this view, SUN2 contributes to the clustering and illegitimate rejoining of multiple DSBs generated by AsiSI[21,22]. However, if and how microtubules also promote

---

[1]Department of Biomedical and Clinical Sciences, Division of Molecular Medicine and Virology, Faculty of Medicine and Health Sciences, Linköping University, Linköping, Sweden. [2]IFOM ETS - The AIRC Institute of Molecular Oncology, Milan, Italy. ✉e-mail: francisca.lottersberger@liu.se

the mobility of one-ended DSBs generated by PARPi in BRCA1-deficient cells has not been investigated.

The mobility of DSBs is also directly regulated by components of the nucleoskeleton. In mammalian cells, nuclear actin is required to mediate DSBs relocalization at nuclear pore complexes (NPC) and/or clustering inside the nucleus[21–27], while in yeast, which undergoes closed mitosis, nuclear microtubules polymerize at DSBs and relocalize them to NPC[28]. In contrast, the nuclear lamin component Lamin-A/C contributes to the stabilization of DSBs[29]. Lamin-A has been shown to locally tether the two broken ends together during NHEJ into a Ku70/Ku80-XRCC4–IFFO1 nucleoskeleton axis[30], but it can also generally constrain chromatin mobility[31,32], possibly by altering the morphology and/or the viscoelastic properties of the nucleus[33,34]. Therefore, if the general viscoelastic properties of the nucleus could affect the microtubule forces promoting DSBs mobility and if the deformability of the nuclear envelope (NE) could be targeted to increase PARPi efficacy in BRCA1-deficient tumors is still unknown.

Here we show that after induction of one-ended DSBs by the PARPi Olaparib in BRCA1-deficient cells, there is a significant increase in NE invagination due, at least in part, to microtubule dynamics. Furthermore, we show that increasing NE invagination either by deletion of the Lamin-A/C encoding gene *Lmna* or by depletion of the main subunit of the serine palmitoyltransferase 1 (SPT1) enzyme increases DSBs mobility and misrepair in *Brca1*-deleted and PARP-inhibited cells. This effect correlates with increased lethality of BRCA1-deficient cells after treatment with Olaparib. Finally, we show that chemical inhibition of SPT1 increases the efficacy of Olaparib in BRCA1-deficient breast and ovarian cancer cells.

## Results

### Microtubules promote DSBs mobility and nuclear invagination after PARP inhibition in BRCA1-deficient cells

We have previously shown that the microtubule stabilizer taxol reduces mobility of dysfunctional telomeres or DSBs induced by γ-radiation[20]. To confirm that dynamic microtubules promote also the mobility of the DSBs generated by PARPi in the absence of BRCA1, we expressed the neutral DSB marker mCherry-53BP1-2[35] in *Brca1*[F/F] MEFs immortalized by CRISPR/Cas9 knockout of *Tp53*. Three days after *Brca1* deletion, we observed some phosphorylation of the DNA damage kinase CHK2 and no accumulation of mCherry-53BP1-2 foci, while treatment with the PARPi olaparib for 6 h caused robust phosphorylation of CHK2 and the formation of numerous (>20) clear mCherry-53BP1-2 foci in about half of the cells (Fig. 1a and Supplementary Fig. 1a–d). Importantly, 6 hours of PARPi treatment in *Brca1*-wild type cells, where DSBs should be repaired by HR, were not enough to trigger CHK2 phosphorylation, although they were enough to induce the formation of few mCherry-53BP1 foci. However, these foci are less bright and less common compared to *Brca1*-deleted cells treated with PARPi (Fig. 1a and Supplementary Fig. 1c, d; median number of foci/cell: 1 versus 20), indicating that the activation of DNA damage response is weaker and/or that the DSBs are repaired more quickly. We then analyzed the mobility of one-ended DSBs in BRCA1-deficient cells treated with PARPi as previously described[20,36]. First, nuclei undergoing large-scale deformations that prevent unambiguous tracking of single foci mobility, were excluded from the analysis (Supplementary Fig. 1e). Then, we calculated the Mean Square Displacement (MSD) and the cumulative distance (CD) of all the traced 53BP1 foci (Fig. 1b, c and Supplementary Fig. 1f). As expected, DSBs induced by PARPi in the absence of BRCA1 roam the nucleus similarly to dysfunctional telomeres and IR-induced DSBs[20,35], with a final MSD of about 0.3 μm² and a median CD of about 2.4 μm after 10 min (Fig. 1b, c and Supplementary Fig. 1f). Furthermore, both taxol, a microtubule stabilizer, and nocodazole, a microtubule depolymerizer, reduced the mobility of DSBs, without affecting the number of excluded nuclei (Fig. 1b, c and Supplementary Fig. 1e, f).

To determine whether the increased mobility was due to microtubules dynamically deforming the nucleus, we visualized the shape of the nuclei with or without DNA damage by immunofluorescence (IF) for Lamin-B1 in fixed samples. Indeed, after deletion of *Brca1* and treatment with PARPi for 6 h, we detected a significant increase in NE invaginations compared to the relatively smooth NE of BRCA1-proficient untreated cells, while no formation of blebs was observed (Supplementary Fig. 1g, h). On the contrary, the treatment with PARPi in BRCA1-proficient cells or the deletion of *Brca1* without PARPi treatment caused only a slight, and not statistically-significant, increase in NE invaginations, when compared to the control cells (Supplementary Fig. 1g, h). Since both PARP inhibition or *Brca1* deletion alone also cause only a mild activation of the DNA damage response (Fig. 1a and Supplementary Fig. 1c, d), these data indicate that the NE shape is altered in response to DNA damage response activation, consistently with a recent study showing that the DNA damage kinases Ataxia telangiectasia mutated (ATM), Ataxia telangiectasia and Rad3 related (ATR) and DNA Protein kinase (DNA-PK) contribute to the formation of Lamin-B-rich tubules in human cells after DSBs induction with etoposide[37].

The formation of lateral invaginations was also confirmed by electron microscopy (EM) (Fig. 1d, yellow panel). Furthermore, EM sections showed also the appearance inside the nucleus of distinct nuclear areas surrounded by an electron-dense material, compatible with transversal sections of NE invaginations (Fig. 1d, blue panel), and recognizable regions that appear separated by a very narrow cytoplasmic channel, which could be compatible with longitudinal sections of NE invaginations (Fig. 1d, magenta panel). Importantly, there was a significant increase in the number of NE abnormalities after *Brca1* deletion and PARPi treatment (Fig. 1d, e). Furthermore, both IF and EM analysis showed that the addition of taxol for one hour prevents the significant accumulation of NE deformations observed in BRCA1-deficient cells treated with PARPi (Fig. 1d, e and Supplementary Fig. 1i), indicating that, in the presence of one-ended DNA breaks, microtubule dynamics plays a major role in NE deformations. However, taxol treatment reduced only partially (and not statistically-significantly) the number of NE abnormalities in *Brca1*-deleted cells treated with PARPi, suggesting that other cytoskeletal or nucleoskeletal components may contribute to NE deformations alongside dynamic microtubules. Surprisingly, nocodazole treatment caused a significant increase in the number of NE invagination even in untreated conditions (Supplementary Fig. 1j), suggesting that the presence of stable microtubule is required to preserve NE morphology and that the formation of NE invaginations per se is not enough to promote DSBs mobility. These data are consistent with recent studies showing the contact between DSBs and NE invagination and the formation of microtubule/LINC-dependent Lamin-B-rich tubules in BRCA1-deficient breast cancer cells treated with etoposide, although in the latter case the formation of tubules was completely abrogated by nocodazole[25,37,38].

### Lamin-A/C but not Lamin-B1 prevents invaginations and one-ended DSB mobility and mis-repair

To determine whether nuclear invaginations or intrinsic nuclear deformability could influence DSB repair in BRCA1-deficient cells, we investigated the effect of alterations in the viscoelastic properties of the nucleus on the mobility and repair of DSBs. To this end, we decided to compromise the nuclear lamin network. In fact, while the sporadic NE invaginations in untreated cells are enriched for both Lamin-A/C and Lamin-B1 (Fig. 2a), Lamin-A is known to be the main contributor to NE stiffness while Lamin-B1 is mostly involved in the preservation of NE integrity[32,34,39–42]. To compare the effect of Lamin-A and Lamin-B1 deficiency on DSBs mobility and mis-repair in PARPi-treated BRCA1-deficient cells, we derived immortalized *Brca1*[F/F] *Lmna*[−/−] MEFs litter mates to *Brca1*[F/F] *Lmna*[+/+] MEFs or we used two specific shRNAs to target *LmnB1* gene expression (Fig. 2b, c). Consistent with previous

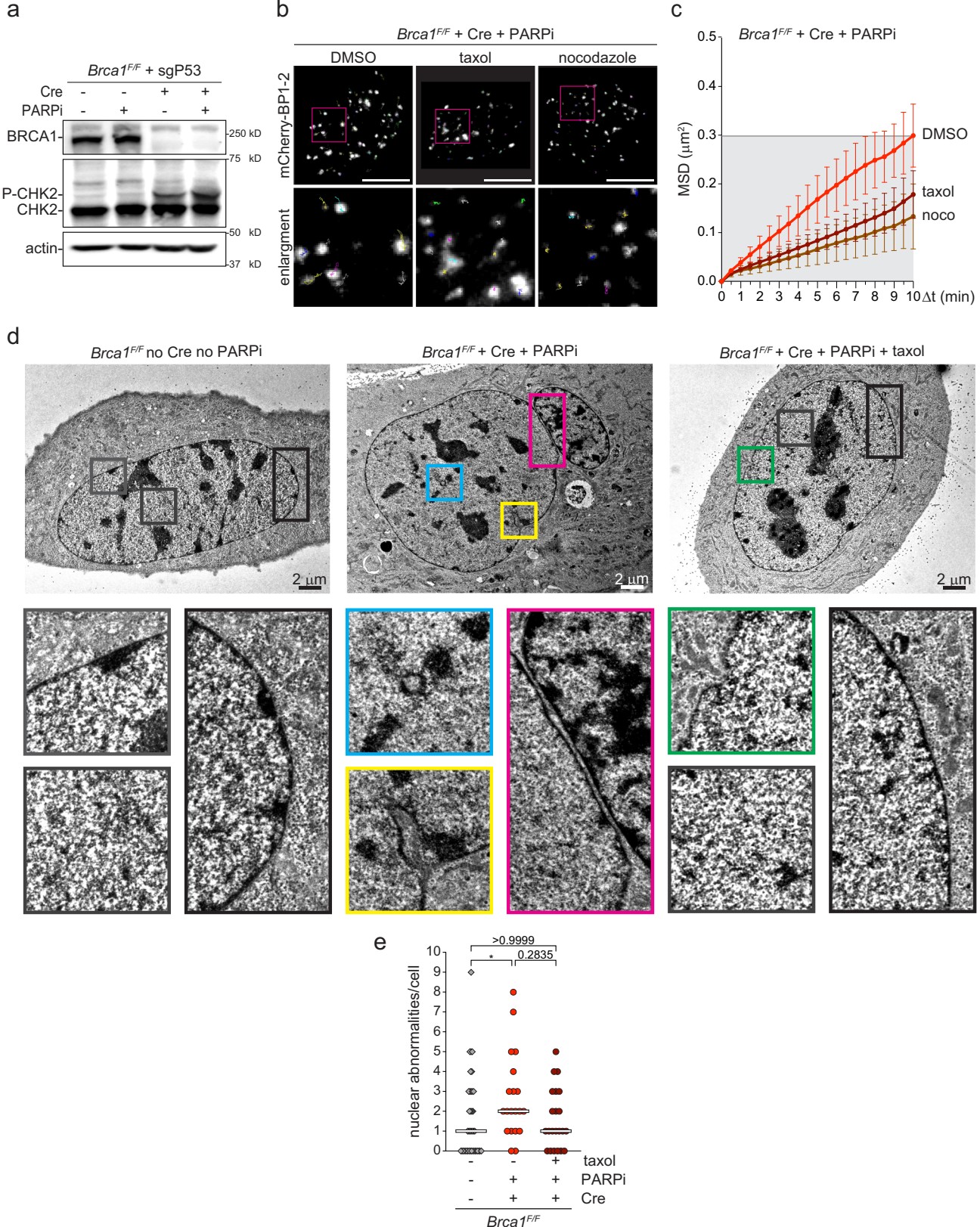

analysis[32,42–44], *Lmna* deletion did not affect cell survival and Lamin A/C-deficient nuclei showed aberrant Lamin-B1 organization and higher levels of invagination, but no blebs/herniations, compared to the control cells (Fig. 2d, e and Supplementary Fig. 2a–c). In contrast, Lamin-B1 depletion caused a significant increase in blebs/herniations, but not in invaginations (Fig. 2d, f), confirming that, while both Lamin-

A and Lamin-B1 control NE structure, their functions are different, and invaginations are specifically prevented by Lamin-A. Furthermore, *Lmna* deletion, but not Lamin-B1 depletion, caused higher mobility of the DSBs induced by PARPi in BRCA1-deficient cells (Fig. 2g–i, and Supplementary Fig. 2d–h), similar to previous data obtained after DSBs induction with etoposide[29,30], indicating that the Lamin-A network, but

**Fig. 1 | Microtubule-dependent nuclear deformations and DSBs mobility.**
**a** Immunoblot for BRCA1, CHK2 and phospho-CHK2 (P-CHK2) in *p53*-deleted (sg-P53) *Brca1^{F/F}* Mouse Embryonic Fibroblasts (MEFs) 72 h after transduction with Hit&Run Cre and/or 6 h treatment with PARPi (0.5 μM). Actin is shown as loading control. **b** Examples of 10 min traces of mCherry-BP1-2 foci in *Brca1^{F/F}* MEFs 72 h after *Brca1* deletion, 6 h after PARPi addition in the absence or presence of taxol (1 h) or nocodazole (2 h). Scale bars: 10 μm. **c** MSD of mCherry-BP1-2 foci in the indicated MEFs as described in (**b**), with SD. Total foci analyzed: 1618 for DMSO, 1056 for taxol, and 1085 for nocodazole from 35, 30, 28 nuclei, respectively from *n* = 3 independent experiment. **d** Representative image of *Brca1^{F/F}* MEFs without any

treatment or 72 h after Cre-mediated deletion of *Brca1*, PARPi treatment for 6 h treatment and/or incubation with taxol for 1 h. Highlighted boxes indicate nuclear deformations as vertical invaginations (blue), light (green) and deep (yellow) lateral invaginations, and longitudinal invaginations (magenta). In black and gray boxes, the control nuclear edges and nuclear interior with no invaginations, respectively. Scale bars: 2 μm. Magnification ×4.25. (**e**) Quantification of nuclear deformations as in (**d**) from 40, 20 and 22 cells for each condition derived from one representative experiment. Statistical analysis by Kruskal–Wallis test for multiple comparisons. (*) *P* < 0.05 (*p* = 0.0381); *P* ≥ 0.05 are not significant. Source data are provided as a Source Data file. See also Supplementary Fig. 1.

not Lamin-B1, prevents DSBs mobility by increasing NE rigidity. This increase in DSB hyper mobility in the absence of Lamin-A was strongly, although not completely, reduced by taxol treatment (Fig. 2g, h and Supplementary Fig. 2d–f), indicating that dynamic microtubules are the main, but perhaps not the only, cytoskeleton force responsible for DSBs mobility in the absence of Lamin-A/C.

Importantly, after removal of BRCA1[45], *Lmna* deletion, but not Lamin-B1 depletion, further reduced cell survival, exacerbated the hypersensitivity to PARPi, and induced a significant increase in the formation of aberrant chromosomes (radials) after PARPi treatment (Fig. 3a–f and Supplementary Fig. 2i,j). Consistent with a previous report showing that Lamin-A/C-deficient MEFs are not hypersensitive to etoposide[30], Lamin-A/C deficiency did not affect the survival to PARPi of BRCA1-proficient MEFs, which can repair DSBs by HR (Fig. 3e), while Lamin-B1 depletion slightly reduced it (Fig. 3f). Finally, the ectopic overexpression of a GFP-tagged human LMNA (GFP-hLMNA) suppressed the deformations (Supplementary Fig. 3a, b), the increased mobility (Supplementary Fig. 3c–f), the aberrant repair (Supplementary Fig. 3g, h), and the reduced survival (Supplementary Fig. 3i, j) of Lamin-A/C-deficient *Brca1*-deleted cells after PARPi treatment. These results indicate that the NE invaginations, caused by the absence of Lamin-A/C, correlate with higher mobility of one-ended DSBs and increased misrepair, while neither of them appears to be affected by other kind of deformations such as NE blebs, as the ones due to Lamin-B1 absence.

## Sphingolipids prevent NE invaginations and one-ended DSB mobility and mis-repair

Lamin-A/C affects several aspects of cell physiology, including chromatin organization and gene regulation[46] and could be directly involved in DSBs repair by stabilizing 53BP1 and the synapsis of the two DNA ends together with Ku70/Ku80[30,47,48]. However, both these functions of Lamin-A/C appear counterintuitive in the context of BRCA1-deficient PARPi-treated cells, where deletion of *53bp1* or *Ku70* prevents radials formation. Furthermore, we did not observe a reduction in 53BP1 levels in the absence of Lamin-A/C (Fig. 2b). Nevertheless, we decided to validate the effect of NE alterations independently of Lamin-A/C deficiency. Since sphingolipids reduce the fluidity and the curvature of cell membranes, including the NE, due to their high bending rigidity[49–52], we decided to target directly the lipids composition of the NE. To this end, we depleted the main subunit of the SPT long chain base subunit 1 (SPT1) with a specific shRNA (Fig. 4a). As expected, SPT1 depletion caused a significant increase in nuclear invaginations but not in blebs, similar to *Lmna* deletion, without affecting survival, Lamin-A/C and Lamin-B1 levels or organization (Fig. 4a–c and Supplementary Fig. 4a, b). Furthermore, it caused a significant increase of DSBs mobility (Fig. 4d, e and Supplementary Fig. 4c, d), radials formations (Fig. 4f, g), and lethality (Fig. 4h, i and Supplementary Fig. 4e) after PARPi treatment in BRCA1-deficient cells. In contrast, in BRCA1-proficient cells, SPT1 depletion did not have any effect on PARPi sensitivity (Fig. 4i). These data confirm that the NE invaginations promote the mis-repair of one-ended DSBs when HR is compromised by increasing their mobility.

## Combined inhibition of SPT1 and PARP increases the lethality of BRCA1-deficient cancer cells

We next investigated the possibility of targeting the NE in cancer cells. To this end we used myriocin, a specific inhibitor of SPT[53]. Treatment with myriocin increased NE invaginations in the BRCA1-deficient p53-deficient breast cancer cell line HCC-1937 and the effect of PARPi in compromising their survival (Fig. 5a–c and Supplementary Fig. 5a, b). However, as SPT1 depletion or *Lmna* deletion in MEFs, myriocin did not affect the PARPi-sensitivity of breast cancer cells that are proficient in BRCA1, such as MCF7 or MDA-MB-231, regardless of their p53 status (Supplementary Fig. 5a–d). Similarly, myriocin increased the lethality of PARPi in the p53-deficient BRCA1-deficient ovarian cancer cell line uWB1.289, but not when the cells were complemented with functional BRCA1 (Fig. 5d–f and Supplementary Fig. 5a, b), indicating that the pharmacological induction of NE invaginations could increase the efficacy of PARPi therapy in BRCA1-deficient cancers. On the contrary, treatment with sub-lethal doses of taxol did not increase the efficacy of PARPi treatment in HCC-1937 and slightly decreased it in uWB1.289 (Supplementary Fig. 5e–h), confirming a requirement for dynamic microtubules in the PARPi-sensitivity of BRCA1-deficient cells.

## Discussion

Here we show that an increase in NE deformability promotes the microtubule-dependent mobility and mis-joining of one-ended DSBs when HR is compromised. Such deformability has to take the form of invaginations and can be induced by microtubules in the physiological context of the cellular response to DSBs, as shown also by others[25,37,38], or by artificially affecting the nuclear Lamin-A/C network or the sphingolipids composition of the NE, as shown here (Fig. 5g). Importantly, dynamic microtubules are still essential to promote DSBs mobility even in the presence of NE invaginations. How microtubules promote DSB mobility and mis-repair is still unclear. We have previously proposed that microtubules could promote mobility by untargeted poking of the nucleus or by direct interaction with DSBs, and that a higher mobility of distant one-ended DSBs would increase their chances to meet and be aberrantly ligated together[20]. The observation that DSB mobility is abolished in nocodazole-treated cells while there are still NE invaginations would now favor the second hypothesis, as would also be supported by several studies showing the recruitment of DSBs or collapsed replication forks to the NE in human cells[25,37,38,54]. However, it is also possible that the NE invaginations observed in the presence of nocodazole are different in nature and/or are more plastic than the ones induced by dynamic microtubules and therefore cannot stimulate mobility. In this scenario, dynamic microtubules could still promote mobility by untargeted poking. Indeed, it has been shown that DSBs can either be recruited at NE by SUN1 to be repaired by HR or to nuclear clusters by SUN2 to undergo NHEJ[21,38], suggesting that the LINC complex, and maybe the microtubules, could play different roles in different repair pathways.

Since NE invaginations are detrimental for cell survival to DSBs in absence of BRCA1, but do not affect cells with intact HR, it is unclear why the cells respond to DNA damage by increasing their formation. It

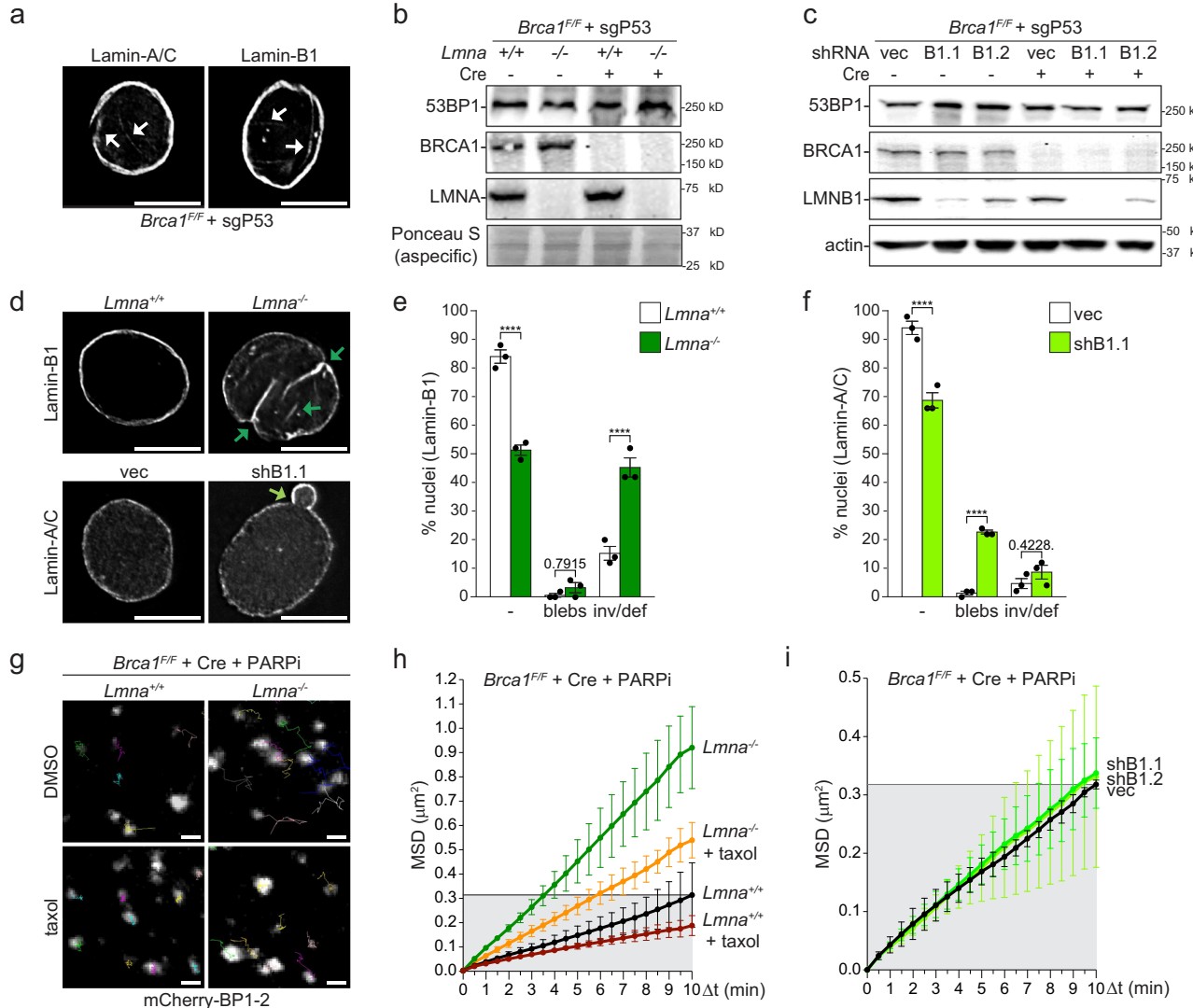

**Fig. 2 | Lamin-A but not Lamin-B1 suppresses DSBs mobility in BRCA1-deficient cells treated with PARPi. a** Representative images of Immunofluorescence (IF) with anti- Lamin-A/C or Lamin-B1 antibodies of *p53*-deleted *Brca1^{F/F}* MEFs. Cells were fixed before immunostaining. Arrowheads indicate nuclear invaginations. Scale bar: 10 μm. **b** Immunoblot for 53BP1, BRCA1 and Lamin-A (LMNA) in *p53*-deleted *Brca1^{F/F} Lmna^{+/+}* or *Brca1^{F/F} Lmna^{-/-}* litter mates MEFs without any treatment or 72 h after *Brca1* deletion with Hit&Run Cre, as indicated. Ponceau S staining is shown as loading control. Immunoblot for 53BP1 was processed in parallel using the same samples. **c** Immunoblot for 53BP1, BRCA1, Lamin-B1 (LMNB1) in *p53*-deleted *Brca1^{F/F}* MEFs transduced with the empty vector control (vec) or two independent shRNAs against (mRNA) *LmnB1* (B1) before and 72 h after Cre-mediated deletion of *Brca1*. Immunoblot for actin is shown as loading control. Immunoblots for 53BP1 and actin were processed in parallel using the same samples as the other two blots. **d** Representative IF images with anti-Lamin-B1 antibodies of *Brca1^{F/F} Lmna^{+/+}* or *Brca1^{F/F} Lmna^{-/-}* MEFs, or with anti-Lamin-A antibodies of *Brca1^{F/F}* MEFs transduced with vec or the shRNA against *LmnB1* as in described in

(**b, c**). Arrowheads indicate nuclear invaginations or blebs, respectively. **e, f** Quantification of normal nuclei and nuclei with blebs or invaginations/deformations in MEFs deleted of *Lmna* (**e**) or depleted of Lamin-B1 (**f**) as indicated and shown in (**d**) for n = 3 independent experiments with mean ± SEM. For each experiment, 50 cells were analyzed for condition. **g** Examples of 10 min traces of mCherry-BP1-2 foci in *Brca1^{F/F} Lmna^{+/+}* or *Brca1^{F/F} Lmna^{-/-}* MEFs 72 h after *Brca1* deletion, 6 h after PARPi addition in the absence or presence of taxol. Scale bar: 1 μm. **h, i** MSD of mCherry-BP1-2 foci in the indicated MEFs as described in (**g**), with SD. Total foci analyzed in (**h**): 1442 for *Lmna^{+/+}*, 1487 for *Lmna^{+/+}* + taxol, 711 for *Lmna^{-/-}* and 1374 for *Lmna^{-/-}* + taxol from 42, 37, 21, 37 nuclei, respectively, from n = 3 independent experiments. Total foci analyzed in (**i**): 998 for vec, 1250 for shB1.1, 959 for shB1.2, from 30, 36, and 27 nuclei, respectively, from n = 3 independent experiments. Statistical analysis by two-way Anova for multiple comparison for (**e, f**). (****) $p < 0.0001$, $P \geqq 0.05$ is not significant. Source data are provided as a Source Data file. See also Supplementary Figs. 2 and 3.

is possible that, as we and others have previously suggested for DSBs mobility, clustering and/or relocalization[20,21,37,38], NE invagination could help the repair of one or few DSBs but would become detrimental in the presence of multiple irreparable DSBs. Further studies will be required to understand how or in which context NE invaginations contribute to proper repair.

It also remains to be elucidated how NE invaginations are generated. Several hypotheses have been proposed, such as DNA-PK activating microtubule dynamics[55], ATM and ATR mediated phosphorylation of

SUN1[37], or the recently discovered ATM-dependent phosphorylation of Lamin-A/C[56,57]. In the context of PARP inhibition in BRCA1-deficient cells, the treatment with taxol reduced the NE invaginations, indicating that microtubules play a major role in their generation. However, the reduction is only partial. While this could be simply due to the limited duration (1 h) of taxol treatment, dictated by its toxicity, it is still possible that other cyto- or nucleoskeleton components could contribute to NE deformations, as nuclear actin or chromatin decompaction[24,58]. It is therefore possible that multiple pathways regulate both the

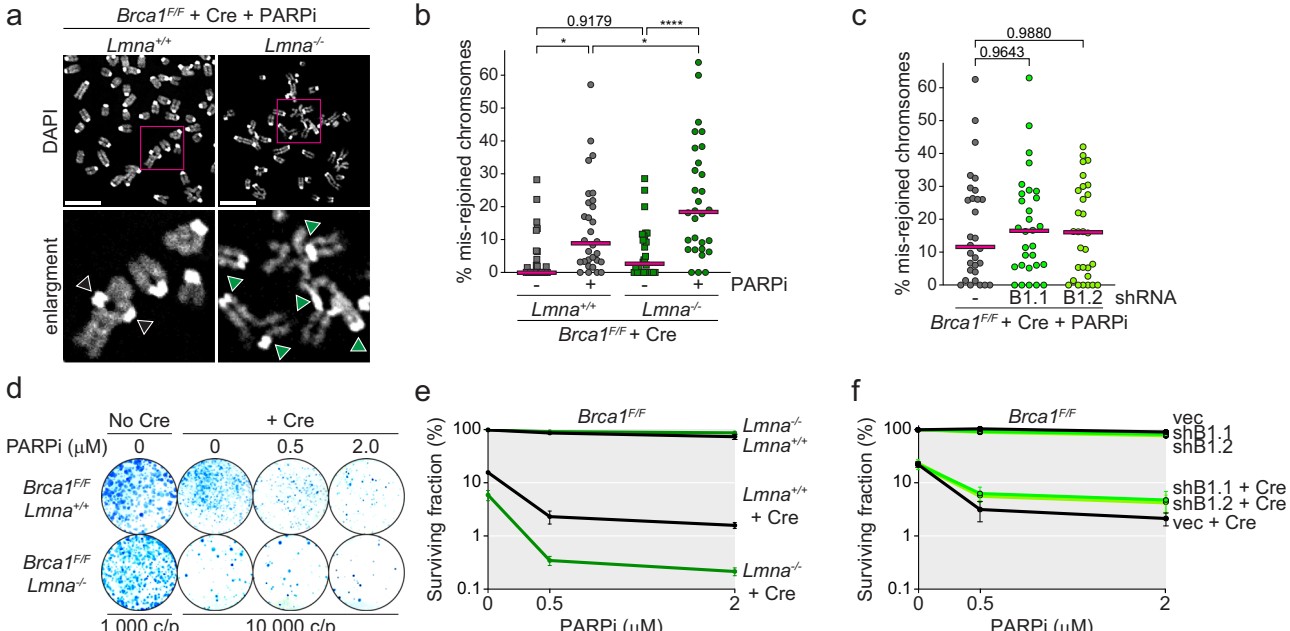

**Fig. 3 | Lamin A but not Lamin B1 suppresses aberrant repair in BRCA1-deficient cells treated with PARPi. a** Representative mis-rejoined chromosomes in *Brca1^{F/F} Lmna^{+/+}* and *Brca1^{F/F} Lmna^{−/−}* 96 h after *Brca1* deletion with Hit&Run Cre and PARPi treatment for 24 h. DNA is visualized with DAPI. Arrowheads indicate centromeres of chromosomes involved in mis-rejoining events. Scale bar: 10 μm. **b, c** Quantification of the percentage of mis-rejoined chromosomes per metaphases in *Brca1^{F/F} Lmna^{+/+}* and *Brca1^{F/F} Lmna^{−/−}* (**b**) or *Brca1^{F/F} Lmna^{+/+}* transduced with vec or two shRNAs against (mRNA) *LmnB1* (B1) (**c**) as shown in (**a**). Each dot represents a metaphase. Bars represent the median for *n* = 3 independent experiments, with 10 metaphases each (30 metaphases in total). Statistical analysis by ordinary one-way ANOVA for multiple comparisons: (*) *P* < 0.05 (0.0167 and 0.0441, respectively); (****) *P* < 0.0001; *P* ≧ 0.05 is not significant. **d** Representative

survival assay of *Brca1^{F/F} Lmna^{+/+}* and *Brca1^{F/F} Lmna^{−/−}* MEFs treated with or without Hit&Run Cre and/or the indicated concentration of PARPi. Cells were treated with PARPi for 24 h before wash. **e** Quantification of survival to the indicated concentrations of PARPi in *Brca1^{F/F} Lmna^{+/+}* and *Brca1^{F/F} Lmna^{−/−}* MEFs before or after Cre-mediated deletion of *Brca1*, as shown in (**d**). Colonies were stained with Methylene blue and their number in the different conditions was normalized over the number of colonies recovered in the untreated plates. Data represents the average and SEMs from *n* = 3 independent experiments. **f** Quantification of survival to the indicated concentrations of PARPi in *Brca1^{F/F} Lmna^{+/+}* MEFs transduced with vec or depleted of Lamin-B1 before or after Cre-mediated deletion of Brca1. Data represents the average and SEMs from *n* = 3 independent experiments. Source data are provided as a Source Data file. See also Supplementary Figs. 2 and 3.

cytoskeleton and the NE to control DSBs mobility and repair, with these pathways varying in activity across different cells and/or in response to specific types of damage.

When NE invaginations are artificially induced, they increase the sensitivity of BRCA1-deficient cancer cells to olaparib, providing a proof of principle for combining the targeting of PARP and NE deformability in cancer therapy. On the contrary, the observed reduction in the PARPi efficacy after treatment with taxol in the BRCA1-deficient ovarian cancer uWB1.289 cell line would argue against the combined treatment of olaparib with paclitaxel, where olaparib has already been shown to not increase the efficacy of treatment with paclitaxel and carboplatinum[59,60]. Finally, since changes in the NE are commonly observed during cancer progression and serve as indices for grade and prognosis[61,62], we would like to suggest that NE deformation can also be used as valuable prognostic marker for PARPi efficacy.

## Methods

### Cell lines, treatments, plasmids, shRNA
*Lmna^{−/−}* mice[63] (009125, the Jackson Lab) was used to derive all the genotypes by standard crosses with *Brca1^{F/F}* mice[64]. Mice were housed and cared for under the Rockefeller University AIACUC protocol 16865-H at the Rockefeller University's Comparative Bioscience Center, which provides animal care according to NIH guidelines. MEFs were isolated at E12.5 embryos and immortalized by electroporation with Cas9 and a sgRNA against *P53* (sequence: 5′-ACACCGTAA-TAGCTCCTGCA-3′). Sex was not assessed. MEFs were cultured in Dulbecco's Modified Eagle Medium (DMEM) (Corning) supplemented

with 15% fetal bovine serum (FBS) (Gibco), non-essential amino acids (Gibco), L-glutamine (Gibco), penicillin/streptomycin (Gibco), 50 μM β-mercaptoethanol (Sigma).

293T/17 [HEK 293T/17] (CRL-11268) and Phoenix ECO cells (CRL-3214) were obtained by ATCC, Rockville, MD, and cultured in DMEM supplemented with 10% HyClone Calf Serum (Cytiva), non-essential amino acids, L-glutamine and penicillin−streptomycin. UWB1.289 (ATCC CRL-2945), UWB1.289 + BRCA1 (ATCC CRL-2946) were obtained from IFOM Human biorepository unit and cultured in 50% RPMI-1640 Medium (Gibco), 50% MEGM (Lonza) made of MEBM basal medium and SingleQuot additives including BPE, hEGF, Insulin and hydrocortisone, supplemented with 10% FBS, L-glutamine and penicillin−streptomycin. HCC1937 (DSMZ ACC 513) were obtained from IFOM Human biorepository unit and cultured in RPMI-1640 supplemented with 10% FBS, L-glutamine and penicillin−streptomycin. MCF7 and MDA-MB-231 (a kind gift from Charlotta Dabrosin, LiU) were cultured in DMEM supplemented with 10% HyClone Calf Serum, non-essential amino acids, L-glutamine and penicillin−streptomycin.

Cre was introduced by 6 infections, 4−12 h apart with the pMMP Hit&Run Cre retrovirus derived from transfected Phoenix cells as previously described[65]. Time point 0 was set at the last Hit&Run Cre infection.

For expression of mCherry-53BP1-2-pWZL[35] or pBabe-puro (Addgene # 1764; kind gift from Jay Morgenstern and Hartmut Land)[66]/pBABE-puro-GFP-hLMNA (GFP-hLMNA; Addgene # 17662; kind gift from Tom Misteli)[67], 20 μg of plasmid DNA was transfected into Phoenix cells using CaPO₄ precipitation as previously described[68]. The retroviral supernatant was used for six infections at 4 h intervals. Cells

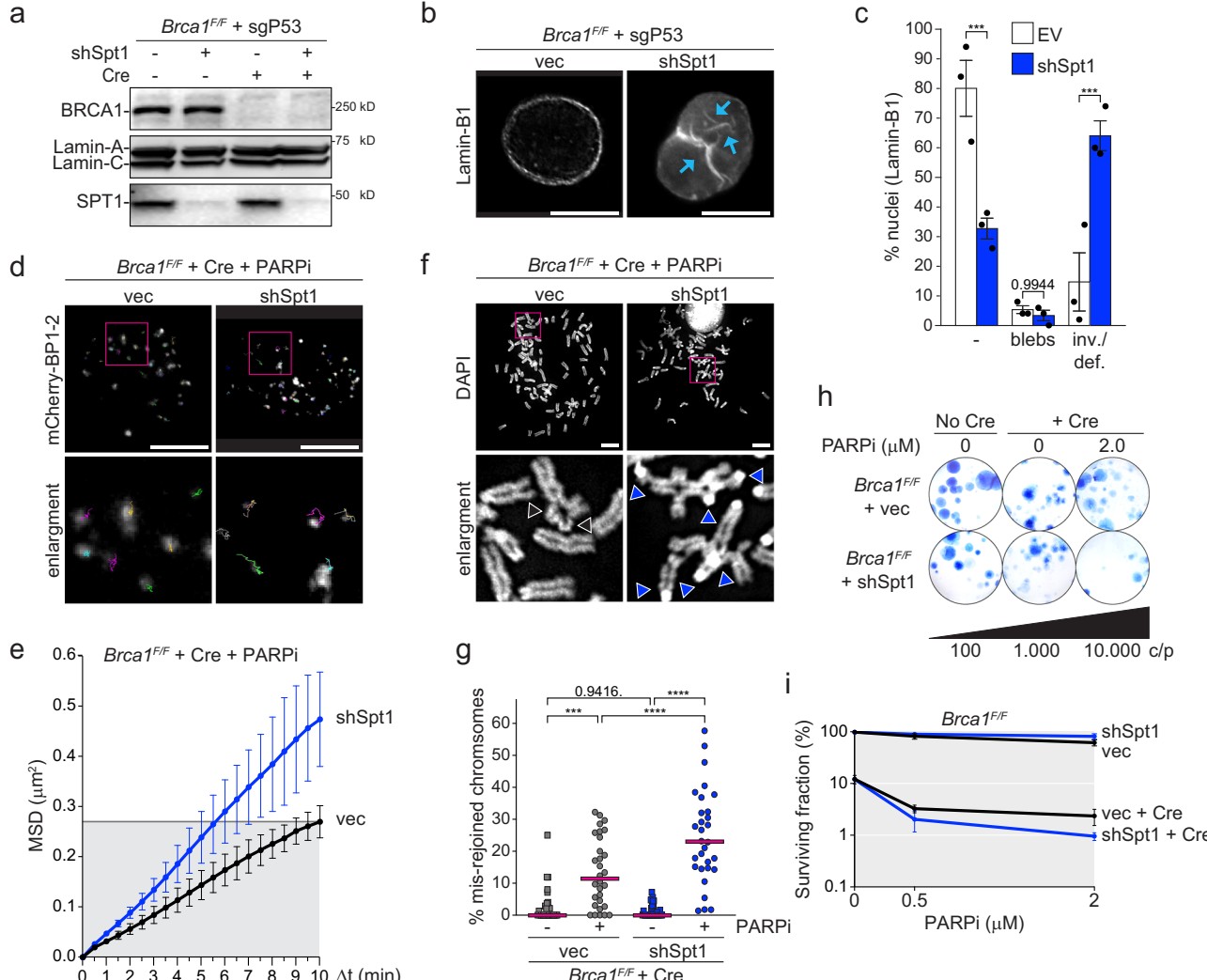

**Fig. 4 | Sphingolipids affects DSBs mobility, DSB misrepair and survival after PARP inhibition in BRCA1-deficient MEFs. a** Immunoblot for BRCA1, LMNA/C and SPT1 in *P53*-deleted *Brca1^F/F* MEFs transduced with the empty vector control (vec) or an shRNA against (mRNA) *Spt1* (shSpt1) before or 72 h after Hit&Run Cre-mediated deletion of *Brca1*. **b** IF for Lamin-B1 of representative nuclei of *Brca1^F/F* MEFs transduced with vec or shSpt1. **c** Quantification of nuclear deformations blebs or invagination/deformations as shown in (**b**). Data are from 150 nuclei from *n* = 3 independent experiments (50 cells/experiment) with mean ± SEM. Statistical analysis by 2way Anova for multiple comparison: (***) *P* < 0.001 (0.0005 for EV and 0.0003 for shSpt1); *P* ≥ 0.05 is not significant. **d** Representative 10 min traces of mCherry-BP1-2 foci in the MEFs transduced with the empty vector or shSpt1 72 h after Cre-mediated deletion of *Brca1* and 6 h treatment with PARPi. **e** MSD of mCherry-BP1-2 foci as shown in (**d**) for *n* = 3 independent experiments, with SD (total foci analyzed: 763 from 24 nuclei for vec and 1201 foci from 32 nuclei for

shSpt1). **f** Representative metaphases showing aberrant mis-rejoined chromosomes in MEFs with or without shSpt1. DNA is visualized with DAPI. Arrows show chromosomes in aberrant structures. **g** Quantification of mis-rejoined chromosomes as shown in (**f**). Each dot represents a metaphase. Bars shows the median for *n* = 3 independent experiment, with 10 metaphases each (total = 30 metaphases per condition). Statistical analysis by ordinary one-way ANOVA for multiple comparisons: (***) *P* < 0.001 (0.0002); (****) *P* < 0.0001; *P* ≥ 0.05 is not significant. **h** Representative survival assay of *Brca1^F/F* MEFs transduced with vec or shSPT1, before or after Cre-mediated deletion of *Brca1*. PARPi was added for 24 h before wash. Colonies are stained with Methylene blue a week later. **i** Quantification of colony formation as shown in (**h**), normalized on MEFs growing without PARPi before *Brca1* deletion. Data represents average and SEM for *n* = 3 independent experiments. All scale bars, 10 µm. Source data are provided as a Source Data file. See also Supplementary Fig. 4.

were selected for 3–5 days in 170 µg/ml hygromycin or 2–4 days in 2 µM puromycin.

Two shRNAs against (mRNA) *LmnB1* (with the following target sequences: shRNA.1: 5′-GGACTTGGAGTTTCGTAAAT-3′; shRNA.2: 5′-AGATAGATGTTGATGGGAAT-3′) were introduced with three infections/day (4 h intervals) over two days using the pSup retroviral vector produced in phoenix ECO cells and infected cells were selected for 2-4 days in puromycin. SPT1 depletion with shRNA in pLKO.1 was performed using the following target sequence: 5′-CCCTGCTCTCAACTACAACAT-3′ (TRCN0000103402, Sigma). The shRNA was introduced with three infections/day (4 h intervals) over

two days using the pLKO.1 lentiviral vector produced in 293T cells and infected cells were selected for 2–4 d in puromycin.

Inhibitors were dissolved in DMSO and used as follows (final concentration): PARPi (olaparib, AZD2281, Selleck chemicals) 0.5–10 µM for 6–72 h; taxol (paclitaxel, Sigma) 0.2–20 µM for 1–24 h; nocodazole (M1404, Sigma) 1 µg/ml for 2 h; and myriocin (M1177, Sigma) 25 µM for 24 h.

For survival assays, MEFs were plated in a 6-well plate in duplicate (10–100 cells/ well for the *Brca1^F/F* No Cre and 500–10000 cells/well, for *Brca1^F/F* +Cre condition). After 24 h, cells were treated with 0.5–2 µM PARPi or DMSO for another 24 h before washing and addition of fresh

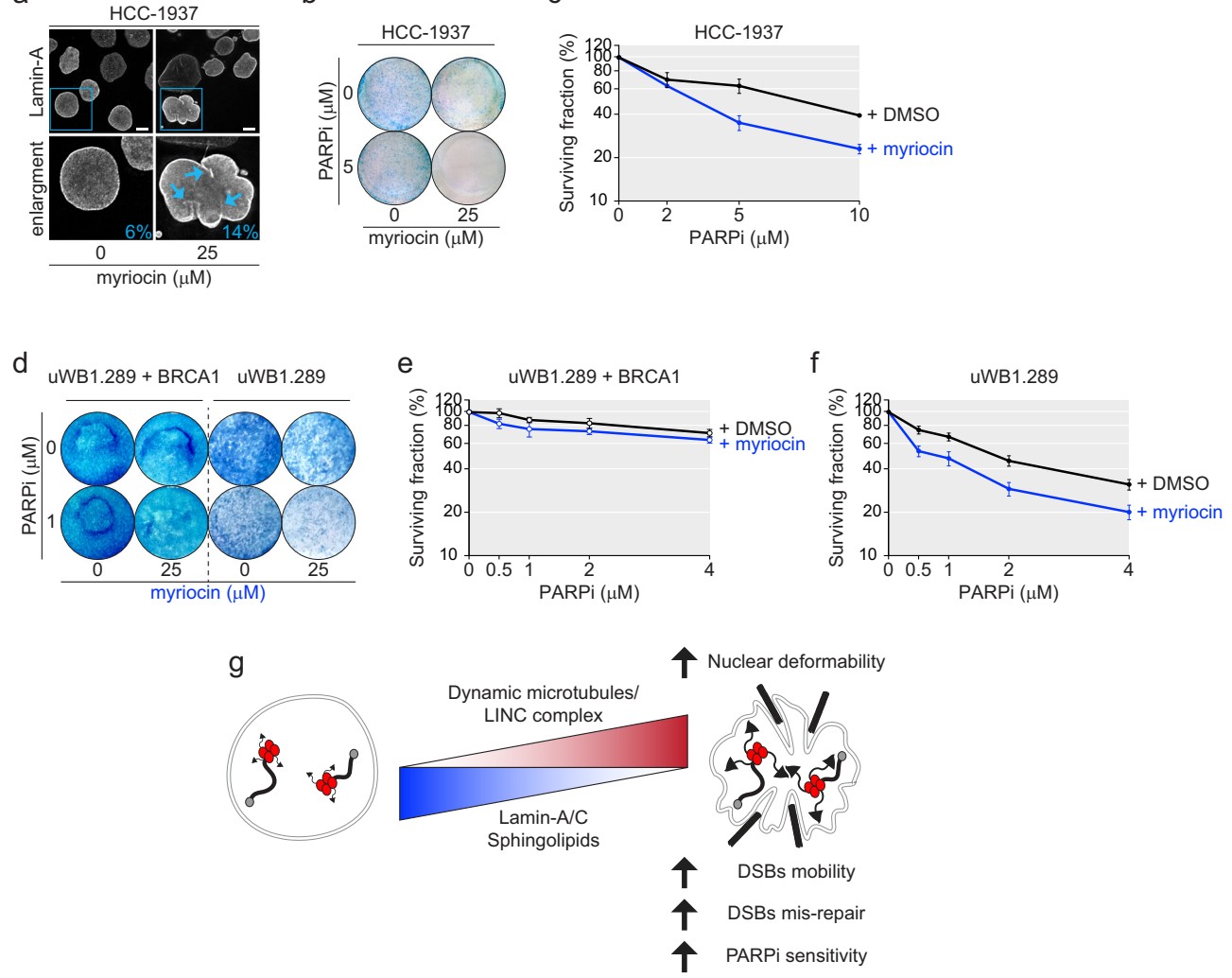

**Fig. 5 | SPT chemical inhibition increases PARPi sensitivity of BRCA-1 deficient human cancer cell lines. a** IF for Lamin-A of representative nuclei of breast cancer HCC-1937 cell line cells treated with DMSO or myriocin for 24 h, with quantification of NE invaginations in one representative experiment. Scale bars, 10 μm. DMSO $n = 63$; myriocin $n = 62$. **b, c** Representative survival assay (**b**) and quantification (**c**) of HCC-1937 cells after exposure for 24 h to myriocin and/or 72 h exposure to PARPi at the indicated concentrations. After removal of the drugs, cells were let grow for a week before harvest. Cells were stained with methylene blue (**b**) or counted after trypsinization (**c**). Quantifications of growth is normalized over the growth without PARPi. Data represents average and SEM for $n = 3$ independent experiments.

Representative survival assay (**d**) and quantification of uWB1.289 ovarian cancer cell line expressing exogenous BRCA1 (**e**) or the original BRCA1-deficient uWB1.289 cell line (**f**) after exposure for 24 h to the indicated concentrations of PARPi and myriocin as described in (**b, c**). Quantifications of growth is obtained independently for uWB1.289 + BRCA1 and uWB1.289. Data represents average and SEM for $n = 5$ independent experiments. **g** Proposed model of how nuclear deformability increases microtubule-dependent DSBs mobility and misrepair, enhancing the sensitivity of BRCA1-deficient cells to PARPi treatment. Source data are provided as a Source Data file. See also Supplementary Fig. 5.

media. After 8-9 days, cells were washed with PBS and colonies were fixed and stained for 2 min in a solution containing 50% methanol, 2% methylene blue and rinsed with water. Colony numbers were determined using wells with 10-40 colonies and the % survival at each PARPi concentration compared to the untreated cells was calculated. Human cancer cell lines were plated in 6 cm plates: HCC-1937, MCF7 and MDA-MB-231 cells at a concentration of $6 \times 10^4$ cells/plate, uWB1.289 and uWB1.289 + BRCA1 cells at a concentration of $8 \times 10^4$ cells/plate. HCC-1937, MCF7 and MDA-MB-231 cells were treated the next day with DMSO or PARPi (2–5–10 μM) alone or in combination with myriocin (25 μM) or taxol (0.2–2 μM) for 24 h before removal of the media. PARPi was then re-added for another 48 h, for a total of 72 h before removal. For uWB1.289 and uWB1.289 + BRCA1 cells, DMSO or PARPi (0.5–1–2–4 μM) and/or myriocin (25 μM) and/or taxol (0.2–2 μM) were added for 24 h before removal of the drugs. Cells were then left to

grow for 6–7 days (HCC-1937, MCF7 and MDA-MB-231) or 5 days (uWB1.289 and uWB1.289 + BRCA1) before being stained or resuspended and counted with a BioRad counter.

**Immunoblotting**

Immunoblotting was performed as previously described[65] with minor modifications. Cells were lysed in 1× Laemmli buffer at $10^7$ cells/ml. The lysate was denatured for 10 min at 95 °C and then sheared with an insulin needle. Lysate equivalent to $25 \times 10^5$ cells/lane was resolved using SDS/PAGE and transferred to a nitrocellulose membrane. Blotting was performed in 5% milk in PBS 1×. The following primary antibodies were used: CHK2 (BD 611570, BD Biosciences); 53BP1 (ab175933, ABCAM), BRCA1 (MAB22101, R&D system); p53 (ab90363, Abcam); SPT1 (ab176906, ABCAM); Lamin-B1 (sc374015, Santacruz); Lamin-A/C (sc518013, Santacruz; 4777S, Cell Signaling).

## Live-cell imaging

Live cell imaging of DSB mobility was performed using mCherry-53BP1-2 (h53BP1, 1220–1711aa)[35] after *Brca1* deletion and PARP inihibition, as previously described[69]. Briefly, $1 \times 10^5$ cells were plated the day after the last infection onto MatTek glass bottom plates and grown for 2 days before imaging. 6 h before the imaging and 72 h after the last Cre infection, PARPi (0.5 μM) was added and, in the last hour before the imaging, the medium was changed into Leibovitz's L-15 medium (Gibco) supplemented with 15% FBS, non-essential amino acids, L-glutamine, penicillin/streptomycin, and PARPi. Imaging was done at 37 °C in an environmental chamber using a DeltaVision RT microscope system (Applied Precision) with a PlanApo 60 × 1.40 NA objective lens (Olympus America, Inc.). Five micrometers Z-stacks at 0.5 μm steps were acquired using SoftWoRx software with 500 ms exposure time, every 30 s over 10 min ($t = 21$ frames) at $1 \times 1$ binning with $1024 \times 1024$ pixels in final size (0.108 μm). Images were deconvolved and 2D-maximum intensity projection images were obtained using SoftWoRx software. Tracking of mCherry-BP1-2 foci was performed with ImageJ software for at least 10 cells per condition. Cells were registered by the StackReg plugin using Rigid Body[70]. Next, particles were detected and tracked for at least 18 of the 20 frames were analyzed using the Mosaic Particle Detector and Tracker plugin[71] with the following parameters: radius = 2-4 pixels; cutoff = 2–4 pixels; percentile = 2–3; link range = 1; displacement = 10 pixels. The x and y coordinates of all the *n* foci detected in each frame were used for the calculation of the coordinates of the Geometrical Center at each time point (GC) (Eq. 1), the Movement of the Geometrical Center (MGC) (Eq. 2), the Difference between the Average Distances of the foci from the geometrical center (ΔAD) (Eq. 3), the percentage of foci moving in the same direction along one of the four quadrants (UR, LR, UL, LL) or in half of the quadrants (LAT, VER, DIA) relatively to the geometrical center and normalized to the coordinates $X_{t=0}$ and $Y_{t=0}$ (Eqs. 4–6), the cumulative distance (CD) (Eq. 7) and the Mean Square Displacement (MDS) (Eqs. 8,9) as:

$$X^{GC}(t) = \frac{1}{n} \times \sum_{i=1}^{n} X_i(t) \text{ and } Y^{GC}(t) = \frac{1}{n} \times \sum_{i=1}^{n} Y_i(t) \quad (1)$$

$$\mathrm{MGC}_{b-a} = \sqrt{(X_{t=b}^{GC} - X_{t=a}^{GC})^2 + (Y_{t=b}^{GC} - Y_{t=a}^{GC})^2} \quad (2)$$

$$\Delta AD_{b-a} = \left( \frac{\sum_{i=1}^{n} \sqrt{(X_{t=b}^i - X_{t=0}^{GC})^2 + (Y_{t=b}^i - Y_{t=0}^{GC})^2}}{n} \right) - \left( \frac{\sum_{i=1}^{n} \sqrt{(X_{t=a}^i - X_{t=0}^{GC})^2 + (Y_{t=a}^i - Y_{t=0}^{GC})^2}}{n} \right) \quad (3)$$

$$LAT(\%) = |(((UR(\%) + LR(\%) \div 100) - 0.5) \div 0.5| \times 100 \quad (4)$$

$$VER(\%) = |(((UR(\%) + UL(\%) \div 100) - 0.5) \div 0.5| \times 100 \quad (5)$$

$$DIA(\%) = |(((UR(\%) + LL(\%) \div 100) - 0.5) \div 0.5| \times 100 \quad (6)$$

$$CD^i = \sum_{t=1}^{20} \sqrt{((x_t^i - x_t^{GC}) - (x_{t-1}^i - x_{t-1}^{GC}))^2 - ((y_t^i - y_t^{GC}) - (y_{t-1}^i - y_{t-1}^{GC}))^2} \quad (7)$$

$$MSD(\Delta t) = \frac{1}{n} \times \sum_{i=1}^{n} D_i(\Delta t), \text{ where} \quad (8)$$

$$D_i(\Delta t) = \sqrt{((x_t^i - x_t^{GC}) - (x_{t-\Delta t}^i - x_{t-\Delta t}^{GC}))^2 - ((y_t^i - y_t^{GC}) - (y_{t-\Delta t}^i - y_{t-\Delta t}^{GC}))^2} \quad (9)$$

Nuclei with one of the three parameters Maximal MGC (MMGC), Maximal ΔAD (MΔAD), or one of UR, LR, UL, LL, LAT, VER, DIA (%) above the arbitrary primary thresholds of 6, 6 or 40%, respectively, or at least two parameters above the arbitrary secondary thresholds of 3, 3 or 30%, respectively, were excluded from the MSD and CD analysis.

## Immunofluorescence

Cells were plated on glass coverslips for 1 day. They were fixed for 10 min in 3% paraformaldehyde/2% sucrose in PBS 1× at RT. After 2 washes in PBS, coverslips were blocked with blocking solution (1 mg/ml BSA, 3% goat serum, 0.1% triton 100×, 1 mM EDTA pH 8.0) for 1 h at RT. Then the coverslips are incubated with primary antibodies overnight at 4 °C. The following primary antibodies were used: 53BP1 (ab175933; Abcam, dilution 1:1000), Lamin-A/C (Cell Signaling, dilution 1:1000), Lamin-B1 (sc374015, Santacruz, dilution 1:750). Cells were washed in PBS three times 5 min each before the addition of the secondary antibody diluted 1:1000 in blocking solution for 45 min at RT. Cells were washed 3x in PBS for 5 min each, with DAPI (1:1000) added to second wash. Cells were left to air dry and mounted with ProLong Gold (Invitrogen).

## Transmission electron microscopy

Cell monolayers attached on a glass coverslip were fixed with 2.5% glutaraldehyde (Polysciences Europe GmbH, Hirschberg an der Bergstrasse, Germany) in 0.1 M phosphate buffer, pH 7.4. Fixed cells were washed in the same buffer and post fixed in 1% Osmium tetroxide (Polysciences Europe GmbH, Hirschberg an der Bergstrasse, Germany). Following washing in phosphate buffer cells were dehydrated in an ethanol series (50%, 70%, 90%, 100%) and anhydrous acetone. Two-steps infiltration first in a mixture of acetone-embedding medium (1:1) and then in 100% embedding media was performed prior to embedding (48 h at 60 °C) in Araldite/Embed 812 embedding media (Electron Microscopy Sciences, P.O. Box 550, 1560 Industry Road, Hatfield, PA 19440). The warm blocks were initially detached from the glass coverslip with help of Liquid nitrogen (LN2), dried out at 60 °C and sectioned using a Leica UC7 ultra microtome (Leica Microsystems GmbH, Vienna, Austria). Ultrathin sections (60-nm thickness) were collected onto formvar-coated copper slot grid, and counter stained with uranyl acetate and lead citrate. Images were taken using 80 kV transmission electron microscope (JEOL JEM1400 Flash, JEOL Ltd., Tokyo, Japan) equipped with a XAROSA camera and RADIUS software (EMISIS GmbH, Münster, Germany).

## Chromosome analysis

For radial chromosome analysis, MEFs were incubated with PARPi (0.5–2 μM) for 24 h before harvest. In the last 2 h, 0.2 μg/ml of colcemid was added. The trypsinized cells were incubated in 10 ml 55 mM KCl at 37 °C for 30 min before being processed. Carefully, 1 ml of cold Carnoy fixative (3:1 ratio of methanol and glacial acetic acid) was added before centrifugation. The pellet was resuspended in 1 ml of cold fixative while vortexing. 10 ml of fresh fixative were added, and samples were stored at 4 °C overnight. Next day, samples were spun and resuspended in 1 ml fixative before being dropped on frosted slides and let dry overnight. When dried, slides were dehydrated in Ethanol before incubation for 10 minutes at 80 °C in hybridization solution (10 mM Tris-HCl pH 7.2, 70% formamide, 0.5% blocking reagent in maleic acid buffer pH 7.5) and telomere PNA probe (Alexa-Fluor 488-OO-(TTAGGG)$_3$, 1:1000 dilution, PNA Bio, not shown) and then for 2 h at RT. Slides were washed in hybridization wash (10 mM Tris-HCl pH 7.2,

70% formamide, 0.1% BSA) twice for 15 min/each and then in PBS three times, 5 min each. To the second wash of 5 min, DAPI (1:750) is added. Slides were dehydrated and air-dried before mounting. Images were acquired using an Elite delta vision microscope.

### Quantification, statistics and reproducibility

Quantification and statistical analysis were performed using Microsoft Excel and GraphPad, respectively, over three or more independent experiments, as indicated. Significance was assessed by calculating the *P* value using unpaired *t*-test or ordinary one-way ANOVA/ Kruskal–Wallis test for multiple comparisons (to compare overall NE deformations, discarded cells, radials formations and survival in two or more than two conditions, respectively), two-way Anova for multiple comparison (to compare different kinds of NE deformations between two conditions), and Mann–Whitney test *t*-test (to compare the distributions of CD). *P* values ≤ 0.05 were considered statistically significant.

All immunoblots have been repeated at least twice on independently collected samples with reproducible results.

### Reporting summary

Further information on research design is available in the Nature Portfolio Reporting Summary linked to this article.

## Data availability

All data supporting the findings of this study are available within the paper and its Supplementary Information. All raw images have been deposited in the Figshare database (https://doi.org/10.6084/m9.figshare.28892306). Source data are provided with this paper.

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

## Acknowledgements

We are extremely grateful to Dr. Titia de Lange for providing critical reagents for this study. We thank Devon White for mouse husbandry. The authors thank Dr. Maria Ntzouni at the Core Facility at the Faculty of Medicine and Health Sciences, Linköping University for electron microscopy sample preparation and electron micrographs acquisition. Dr. Charlotta Dabrosin is thanked for sharing MCF7 and MDA-MB-231 cell lines. This work was supported by grants to F.L. from Cancerfonden (21 1732 Pj), Vetenskapsrådet (2021-02788) and Knut and Alice Wallenberg Foundation. Y.D. is supported by Associazione Italiana per la Ricerca sul Cancro, AIRC IG 28954.

## Author contributions

E.F. performed and analyzed all the experiments. A.d.S. helped with revision. A.P. developed the mobility analysis software with F.L. and E.F. F.L. conceived the study with E.F., provided supervision with Y.D., procured funding and wrote the manuscript with contributions from all authors.

## Funding

## Competing interests

The authors declare no competing interests.
