## [Transparent Peer Review file · Nature Communications]

Nuclear deformability increases PARPi sensitivity in BRCA1-deficient cells by increasing microtubule-dependent DNA break mobility

Corresponding Author: Dr Francisca Lottersberger

Version 0:

Reviewer comments:

Reviewer #1

(Remarks to the Author)

In their manuscript with the title “Nuclear deformations increase PARPi sensitivity in BRCA1-deficient cells by altering DNA break mobility”, Faustini and colleagues further investigate in the contribution of microtubules and the nuclear envelope structure in PARPi- induced DSB repair in BRCA1-deficient cells.

They find increased occurrence of nuclear envelope invaginations in PARPi-treated BRCA1-deficient cells, at least partially dependent on microtubules.

Upon deletion of LMNA/C as well as depletion of the serine palmitoyltransferase multisubunit enzyme (SPTL1), which inhibits sphingolipid formation, the authors scored increased DSB mobility, nuclear invagination, genome instability and a decrease in survival in PARPi-treated BRCA-deficient MEFs.

In line with their experiments in MEFs, Faustini and colleagues observe an augmented sensitivity of BRCA-deficient human cancer cells upon PARP- and sphingolipid- inhibition. This appears particularly interesting for BRCA-deficient cancer treatment as BRCA-proficient cells do not display this phenotype upon the combination-treatment.

The study presented here provides interesting new data in the intersection field of nuclear structure and DNA repair, in particular as a result of PARP-inhibition in BRCA-deficient cells. Although the manuscript cannot convincingly help to understand the contribution of microtubules or nuclear structure to DSB-mobility and -repair, investigations regarding nuclear envelope composition and its manipulation in the context of PARPi-treatment provide interesting novel insights, especially in the context of BRCA-deficient cancer therapy.

Comments:

1. The observation of decreased DSB mobility upon treatment with nocodazole or taxol (Fig. 1b/c) gave rise to the speculation “...whether the increased mobility was due to microtubule directly poking the nucleus...” (line 121). This is conceptually counter-intuitive because DSB mobility is reduced in both conditions, where microtubules are stabilized or destabilized.

As taxol stabilizes microtubules, and taxol-treatment reduces nuclear invaginations (Fig. 1 d/e), the authors should explain how they come to the following conclusion “These data are consistent with the recent identification of microtubule/LINC-dependent NE poking” (line 129).

In the same context, what is the effect of nocodazole-treatment on nuclear structure in the conditions used in Fig. 1d/e and the effect of nocodazole/taxol-treatment on nuclear structure in the conditions used in Extended Data Fig. 1e/f?

What is the effect of PARPi on the nuclear structure of BRCA-proficient cells (-cre/+PARPi) in the conditions used in Fig. 1d/e and used in Extended Data Fig. 1e/f?

Please further elaborate on the role of microtubules in DSB mobility and -repair in the discussion part. What is the proposed mechanism?

2. Please show the effect of myriocin on DSB mobility in human cancer cell lines (Fig. 5).

3. Analysis of human cancer cell survival in Fig. 5: Only one PARPi concentration and a different graphical representation is used (bar diagram, linear y-axis). For better comparison and consistency, I suggest using 0.5 and 2 µg PARPi also in human cancer cell lines and display results in the same graphical representation as in the previous figures for survival curves.

Minor points:

1. line 86, the term mobility is used twice: „The mobility of DSB mobility “
2. line 156, typo: “ Improtantly,...“
3. Figure 3e: graphs as mislabelled – the lowest curve (green) must be LMNA/C -/- and the black curve on top LMNA/C +/-
4. Extended Data Fig. 3i: Error bars are missing for the hLMNA+cre curve.
5. Figure 3e: LMNA/C-deficiency already has an impact on cell survival upon BRCA-loss. Please show relative survival of the respective cell lines upon PARPi (as % to untreated) to allow a better comparison of the PARPi-effect on LMNA/C -/- cells to the LMNA/C +/- counterparts.
6. Line 173: “Furthermore, we did not observe any reduction in 53BP1 levels in the absence of LMNA (Extended Data Fig. 3a)”. Extended Data Fig. 3a does not show a comparison of LMNA/C -/- and LMNA/C +/- cells. Either this statement should be re-phrased or correct western blots need to be shown.
7. Figure 5a, please provide quantification

Reviewer #2

(Remarks to the Author)

The manuscript by Faustini et al. investigates the involvement of microtubules, nuclear lamin proteins (Lamin A/C and Lamin B1), and sphingolipids in the repair of DNA Double Strand Breaks (DSBs), particularly in BRCA1-deficient cells treated with PARP inhibitors (PARPi). They demonstrate that increase in nuclear DSB mobility and envelope (NE) invaginations induced by microtubule dynamics and LaminA/C deficiency correlates with enhanced PARPi cytotoxicity in BRCA1-deficient cells. Additionally, depletion of SPTLC1, a subunit of the sphingolipid synthesis complex, increases NE invaginations and DSB mobility, similar to Lamin A/C depletion. Targeting sphingolipids with myriocin enhances PARPi efficacy in BRCA1-deficient cancer cells without affecting BRCA1-proficient cells. These findings suggest that NE remodeling, particularly through increased invaginations, could serve as a potential therapeutic strategy to enhance PARPi therapy efficacy in BRCA1-deficient cancers.

In general, this work is of high quality, the manuscript is well-written, addressing questions of great interest to researchers in the DNA repair and nuclear organization fields, with strong implication for cancer treatment.

This study nicely parallels a recently published study (Shokrollahi et al. NSMB) describing NE invaginations referred to as dsbNETs, which are likely identical to the NE invaginations observed in the present study (induced by microtubules and enriched in Lamin B1 in BRCA1-deficient cells treated with Olaparib). Affecting these dsbNETs (through KIF5B depletion) decreases sensitivity to PARPi and reduces the ability of PARPi to induce toxic misjoined chromosomes in BRCA1-deficient cells. Conversely, disruption of dsbNETs decreased BRCA1-deficient cells growth and tumorigenicity in mouse xenografts in absence of PARPi treatment. Do the author have an explanation for this opposite effect in presence or absence of PARPi?

The present study confirms that NE invaginations (dsbNETs) increase the PARPi sensitivity of BRCA1-deficient cells. Additionally it shows that increasing NE invaginations upon treatment with myriocin, a FDA approved drug, increases PARPi sensitivity of BRCA1-deficient cells, providing a proof of principle for cancer treatment.

Additionally, the authors show that taxol reduces both DSB dynamics and NE invaginations. However the authors did not test the effect of taxol on PARPi efficacy in BRCA1-deficient cells. The results of this experiment would be particularly relevant, given that taxol is one of the most important anticancer drugs, widely used in clinics for the treatment of ovarian and breast cancer. In this regard, authors should discuss published studies combining taxol (paclitaxel) and PARPi (olaparib) such as the work by Faschling et al. 2021 published in the Annals of Oncology and Zhang et al. 2022 in Am J Transl Res.

Minor concerns:

- Some figures are missing statistics (extended data Fig1. b, Fig1. e difference between BRCA1F/F + PARPi -Taxol vs BRCA1F/F + PARPi +Taxol).

- The conclusion on the top of p7 “These results indicate that the NE invaginations caused by the absence of Lamin A/C increase the mobility of one-ended DSBs, therefore increasing their misrepair” is too strong. Considering the pleiotropic effects of LMNA/C depletion (stabilization of 53BP1 or effect on DSB synapsis), NE invagination may not be causal. The authors should state that they observe a correlation.

- The authors state that Lamin B1 organization is not perturbed upon shSPTLC1; however, they observed increased NE invaginations which is a phenotype typically observed upon LMNB1 overexpression. The authors should thus check LaminB1 level upon shSPTLC1 by western blots or quantification of IFs.

- Extended data Fig.1: Panel d is not well referenced in the figure legend.

I would like to emphasize the clarity of this manuscript and the high quality of the data presented. Addressing the effect of taxol on PARPi efficacy in BRCA1-deficient cells will make this manuscript a perfect fit for publication in Nature Communications.

Reviewer #3

(Remarks to the Author)

In this manuscript, the authors suggest that nuclear deformations affect the mobility of DNA breaks. Increased DNA breaks' mobility is thought to increase sensitivity to the inhibition of PARP in cells that have lost BRCA1. The results, as presented, are hard to interpret and do not support the main conclusions of the manuscript. A major flaw with the experimental approach is that targeting the levels of the nuclear lamins or lipid synthesis will disrupt every aspect of cell physiology and have many pleiotropic effects. Both nuclear lamina and lipid synthesis are essential for viability. In addition, the authors overinterpret some of the results to draw hypotheses about the properties of the nuclear envelope. Without direct assays determining the properties of the nuclear membrane, it is unclear what the images of the nucleus obtained by microscopy mean. Although the data presented is exciting and, with the proper controls, the authors could establish a link between the nuclear envelope (NE) and DNA breaks, the main conclusions of the manuscript are not supported by the experimental approaches and results.

The manuscript, as presented, is hard to read and needs to be revised. It could be improved by addressing unclear terminology, overinterpreting the data, unclear labeling of figures, and spelling and grammatical mistakes.

Line 57: What are NE remodeling drugs?

In the introduction, the authors say that it is unknown whether microtubules (MT) promote the mobility of DSB in BRCA1-deficient cells (line 84). This statement contradicts the previous statement in the abstract, saying MT promotes the mobility of DNA DSBs (line 47).

It is unclear what the viscoelastic properties of the NE mean (line 137). How do viscoelastic properties promote and/or counteract MT forces? How can the properties of the NE be targeted? How can changing the properties of the NE affect the efficacy of drugs that inhibit PARP?

How does targeting of p53 immortalize MEF? This approach is a significant issue in the manuscript. Loss of p53 is not considered in the interpretations of the results. Also, MEFs without p53 tend to become tetraploid and may be genomically unstable.

In Fig. 1a, images of control cells are not shown: BRCA1 F/F – Cre with or without PARPi. Nuclear staining would be helpful in visualizing the nucleus in Fig. 1b.

What is the data that shows highly distorted nuclei?

In extended Fig. 1c, what are the discarded nuclei? What are the excluded nuclei? There is no figure legend for extended Fig. 1d.

The cumulative distance differences are minimal. Most data points for all three conditions lie between 1 and 3 μm .

The effects on nuclear morphology seem minimal in Figure 1d. Also, the analyses of the nuclear morphology of -Cre +PARPi, +taxol alone, or no Cre +PARPi + taxol are not shown.

What is the evidence that MT cause nuclear invaginations? The data in Fig 1e shows a change from 1 to 2 nuclear abnormalities per cell. Is this change physiologically relevant?

It is unclear how, if, or what does it mean MT directly poking the nucleus. What are the data showing MT poking NE?

The signal of lamin B1 in Ext Fig. 1e does not represent NE invaginations. Are these images from confocal microscopy?

It is unclear what is presented in Ext. Fig 1f. What does "-" mean? Normal? Control experiments are missing. +Cre no PARPi or PARPi alone.

Figure 1d: how can you tell a vertical invagination from electron microscopy? What is the separation of nuclear areas? The middle panel shows a micronucleus. What data shows that taxol reduces the separation of nuclear areas?

The data presented in Figure 1 does not strongly support the conclusion that microtubule dynamics promote NE invaginations and mediate DSB mobility and misrepair.

In Figure 2a, it is unclear how NE invaginations are defined.

Experiments after deleting lamin A/C are hard to interpret since lamin A/C is an essential gene. Loss of nuclear lamina will affect every aspect of cell physiology, including transcription, nuclear pores, cell division, and more.

Taxol reduces MSD with or without LMNA, which suggests that MT's effect is independent of LMNA. The data presented in this manuscript do not support the conclusion that Lamin A counteracts the forces generated on the NE by MT.

In Figure 3a, do the arrowheads point to translocations? How can one tell what a misrejoined chromosome is from these images?

Figure 3e shows that PARPi kills 99% of cells with or without lamin A/C. Also, Cre alone reduces viability by 90%, indicating that BRCA1 deletion makes cells very sick.

Data in Ext Fig 3g shows that re-expression of LMNA minimally suppresses % of mis-rejoined chromosomes.

It is unclear how sphingolipids reduce the fluidity and the curvature of cell membranes. SPTLC1 is an essential gene, and lipid synthesis is essential for viability (<https://depmap.org/portal/gene/SPTLC1?tab=overview>).

The results presented in Figure 5 are expected. The combination of PARPi and myriocin is more toxic than either condition alone. The analysis of a few cell lines and one drug concentration is insufficient to draw general conclusions about cancer and SPT or PARP inhibition.

Version 1:

Reviewer comments:

Reviewer #1

(Remarks to the Author)

Faustini and co-workers have added substantial amount of work to improve the manuscript and to address reviewers concerns.

I appreciate their effort and the improved quality of the presented work.

After addressing the two following minor points, I see this work ready for publication in Nature Communications:

1. In line 121 of the revised manuscript, authors state:

"The fraction of 53BP1-positive cells, however, was reduced from 50% to 35% compared to the one observed in BRCA1-deleted cells (Fig. 1a and Extended Data Fig. 1c,d)."

In the data provided in the respective figures, no percentages are given. The graph in Extended Data Fig 1d shows the "number of 53BP1 foci/nucleus".

Please re-phrase or provide missing data.

2. I asked the authors to correct labeling of their survival curves in Figure 3a (minor point 3).

Surprisingly, I still find Figure 3a mis-labeled in the revised manuscript. Please change accordingly.

Reviewer #2

(Remarks to the Author)

The authors have addressed all my concerns in the revised manuscript

Reviewer #3

(Remarks to the Author)

The revised manuscript clarifies many of the confusing points I raised before and is now easier to understand. However, several issues remain to be addressed, and the data provides correlative results to support the main conclusions of the

manuscript.

Almost every western blot (WB) in the manuscript is missing loading controls and molecular markers:

WB of Figure 1a is missing loading control. Levels of Chk2 are variable.

WB in Figure 2b and 2c are missing loading controls. Lamin A/C should be 2 bands.

WB in Ext. Fig. 3a is missing loading controls. The bands are for LMNA/C or LMNA/C-GFP? No MW markers are shown.

WB in Ext. Fig 3j, no loading control.

Figure 4a, WB shows loading control Lamin A/C. There are 2 bands as expected. These two bands are not seen in the manuscript in other westerns of Lamin A/C.

In line 121, 53BP1 positive cells was reduced from 50% to 35% compared to the one observed in BRCA1-deleted cells. Neither Fig. 1a nor Extended Fig. 1c, d show these numbers.

The effects of taxol or nocodazole in DSB mobility in WT +PARPi are not shown.

In Ext Fig.1h, PARPi treatment did not cause a significant increase in NE invaginations in BRCA1+ cells or BRCA1- cells. The comparisons show no significance. There is a significant effect on BRCA1+ /PARPi and BRCA1- /PARPi (red bars); these results suggest that BRCA1 deletion in the context of PARPi causes changes in the NE. PARPi treatment alone does not cause NE invaginations. These results are confusing as to what exactly is causing NE defects.

The EM images show minimal changes in the nuclear envelope morphology. The middle image shows a micronucleus. The quantification in Fig 1e shows that PARPi and BRCA1 deletion increases the nuclear abnormalities from 1 to 2 per cell. It is unclear whether this effect is physiologically relevant and whether or how 1 more nuclear invagination per nucleus could account for changes in DSB mobility in the whole nucleus.

Line 149-150: The taxol effect in Ext Figure 1i shows no significance, contradicting the text.

Nocodazole does not increase NE invaginations in PARPi/BRCA1- cells (Ext Fig 1j last two bars) but reduces mobility (Ext fig 1f). These data contradict the main point of the manuscript that NE inv causes DNA mobility.

Line 155: "NE invaginations is not enough to promote mobility". The paper is confusing because it is unclear what the relationship between deformability and nuclear envelope invaginations and nuclear blebs is.

Figure 2c points to an effect of BRCA1 levels upon knockdown on lamin B.

The MSD of LMNA-/- treated with taxol is greater than control cells (Figure 2h). This indicates that microtubules (MT) are not the major force for DSB mobility. Because taxol reduces DSB mobility by a similar extent (50-60% of the untreated) with or without Lamin A the data in Figure 2h argues that DSB dependence on MT is independent of Lamin A.

The comparison of the data in Ext Figure 2f vs. 2h shows control cells cumulative distance of 2.367 (2f) and 2.531 (2h). That's a difference of 0.164 between experiments. The difference between control and taxol is less than that (2.367- 2.254 = 0.113), indicating that the effects of taxol are within the error of biological replicates.

Data in Ext Figure 2e show that about 80% of nuclei with LMNA-/- are excluded. This suggests that the 20% of cells analyzed may not represent the major physiological consequences of LMNA depletion and that the differences in DSB mobility are minor phenotypes in these cells.

Line 187 states that Lamin B depletion does not affect viability. Figure 3f shows that shLB1 + Cre has significantly lower viability than -Cre.

The results describing the effects of sphingolipid inhibition on DSB remain correlative observations and require further investigation. Without assessing the effects of sphingolipid inhibition on the biophysical properties of the nuclear membrane, these results remain speculative.

Version 2:

Reviewer comments:

Reviewer #3

(Remarks to the Author)

The authors have addressed most of my comments, and the manuscript has significantly improved.

A few comments:

Almost every image in the manuscript is missing scale bars.

The significance of the data in Figure 1e remains unclear. One more invagination per cell and a few EM images may surprise the readership as significant evidence to support the hypothesis postulated in the paper.

It is unclear what the nuclear gap is. Ext fig 2c?

Fluorescent microscopy of cells expressing GFP-lamin A showing the rescue of nuclear invaginations may strengthen the results in Figure 3.

REVIEWER COMMENTS

Reviewer #1 (Remarks to the Author):

In their manuscript with the title “Nuclear deformations increase PARPi sensitivity in BRCA1-deficient cells by altering DNA break mobility”, Faustini and colleagues further investigate in the contribution of microtubules and the nuclear envelope structure in PARPi- induced DSB repair in BRCA1-deficient cells. They find increased occurrence of nuclear envelope invaginations in PARPi-treated BRCA1-deficient cells, at least partially dependent on microtubules.

Upon deletion of LMNA/C as well as depletion of the serine palmitoyltransferase multisubunit enzyme (SPTL1), which inhibits sphingolipid formation, the authors scored increased DSB mobility, nuclear invagination, genome instability and a decrease in survival in PARPi-treated BRCA-deficient MEFs.

In line with their experiments in MEFs, Faustini and colleagues observe an augmented sensitivity of BRCA-deficient human cancer cells upon PARP- and sphingolipid- inhibition. This appears particularly interesting for BRCA-deficient cancer treatment as BRCA-proficient cells do not display this phenotype upon the combination-treatment.

The study presented here provides interesting new data in the intersection field of nuclear structure and DNA repair, in particular as a result of PARP-inhibition in BRCA-deficient cells. Although the manuscript cannot convincingly help to understand the contribution of microtubules or nuclear structure to DSB-mobility and -repair, investigations regarding nuclear envelope composition and its manipulation in the context of PARPi-treatment provide interesting novel insights, especially in the context of BRCA-deficient cancer therapy.

We thank the reviewer for the time dedicated to our manuscript and for their positive summary of our work.

Comments:

1. The observation of decreased DSB mobility upon treatment with nocodazole or taxol (Fig. 1b/c) gave rise to the speculation “...whether the increased mobility was due to microtubule directly poking the nucleus...” (line 121). This is conceptually counter-intuitive because DSB mobility is reduced in both conditions, where microtubules are stabilized or de-stabilized.

As taxol stabilizes microtubules, and taxol-treatment reduces nuclear invaginations (Fig. 1 d/e), the authors should explain how they come to the following conclusion “These data are consistent with the recent identification of microtubule/LINC-dependent NE poking” (line 129).

We agree that it appears counter-intuitive that taxol and nocodazole - which affect the microtubule stability in the opposite ways - have the same effect on (some) microtubule-mediated processes. However, as in this and other experiments of DSBs mobility (as for dysfunctional telomeres, Lotterberger et al., 2015), this phenotype can be explained by the requirement for dynamic polymerization and depolymerization, rather than just the stable presence of microtubules. It is the dynamic nature of microtubules that promotes DSBs mobility. In order to better clarify this point, we have replaced “diectly poking” with “dynamically deforming” in the text.

In the same context, what is the effect of nocodazole-treatment on nuclear structure in the conditions used in Fig. 1d/e and the effect of nocodazole/taxol-treatment on nuclear structure in the conditions used in Extended Data Fig. 1e/f?

Following the questions raised by the reviewer, we investigated more in details the effect of taxol and nocodazole in the accumulation of NE invaginations. IF analysis shows that the taxol treatment did not affect nuclear structure in wt cells, while it reduced the levels of invagination in BRCA1-deficient cells treated with PARPi (new Extended Data Fig.1i). This result is consistent with the the EM data shown in Fig. 1d,e.

Nocodazole treatment on the other hand, lead to a substantial increase in nuclear deformations even in otherwise untreated cells (new Extended Data Fig.1j and below). These defromation appeared as

“collapsed” nuclei and were independent of DNA damage. We suspect that they may be due to a reinforcement of the actomyosin cytoskeleton due to the complete removal of microtubules, which could then cause NE deformations. However, we cannot rule out other possibilities and further work will be required in the future to better explain this phenotype.

Nuclear deformations of Brca1-deficient cells treated with the PARPi, did not change significantly after nocodazole treatment (new Extended Data Fig.1j). The interpretation of this result however, is complicated by the unexpected effect of nocodazole treatment in wt cells.

While the data with taxol support the conclusion that dynamic microtubules are required for NE invagination, these new data with nocodazole, combined with the mobility assay, support the view that NE invaginations may arise through different mechanisms, some of which may not necessary affect the mobility, while dynamic microtubules are essential for mobility. We have modified the text and the discussion accordingly and added a final summary (new Extended Data Fig.5g)

What is the effect of PARPi on the nuclear structure of BRCA-proficient cells (-cre/+PARPi) in in the conditions used in Fig. 1d/e and used in Extended Data Fig. 1e/f?

In the revised version of the manuscript, we have included the IF analysis of NE structure in BRCA1-proficient cells treated with the PARPi and in BRCA1-deficient cells without PARPi treatment (new Extended Data Fig.1h). In these experiemnts, we observed only a slight, and not statistically significant, increase in the invaginations after either BRCA1 deletion or PARPi treatment alone, compared to the untreated cells. Only the deletion of BRCA1 combined with PARPi treatment increases the invagination significantly, consistent with the accumulation of DNA damage.

Please further elaborate on the role of microtubules in DSB mobility and -repair in the discussion part. What is the proposed mechanism?

We have now added the following paragraph in the discussion to present two possible models: “We have previously proposed that microtubules could promote mobility by untargeted poking of the nucleus or by direct interaction with DSBs, and that a higher mobility of distant one-ended DSBs would increase their chances to meet and be aberrantly ligated together (20). The observation that DSB mobility is abolished in nocodazole-treated cells while there are still NE invaginations would now favor the second hypothesis, which would also be supported by several studies showing the recruitment of DSBs or collapsed replication forks to the NE in human cells (25,37,38,55). However, it is also possible that the NE invaginations observed in the presence of nocodazole are different in nature and/or are more plastic than the ones induced by dynamic microtubules and therefore cannot promote mobility. In this scenario, dynamic microtubules could still promote mobility by untargeted poking. Indeed, it has been shown that DSBs can either be recruited at the NE by SUN1 to be repaired by HR or to nuclear clusters by SUN2 to

undergo NHEJ, suggesting that the LINC complex, and maybe the microtubules, could play different roles in different repair pathways.”

2. Please show the effect of myriocin on DSB mobility in human cancer cell lines (Fig. 5).

Our model predicts that myriocin treatment would work by increasing the relative mobility of DSBs also in human cancer cell lines, therefore we agree with the reviewer that these experiments could provide an additional test for our hypothesis. However, we would like to point out that the DSBs mobility assay in 2D using DeltaVision and mcherry-53BP1 is developed and implemented for Mouse Embryonic Fibroblasts, which have very flat nuclei, divide quite rapidly (16-18hr) and where there is a very low background of 53BP1 foci that clearly accumulate after about 6 hours of PARPi treatment. The human cell lines we used for the sensitivity assay are more spherical, divide more slowly (almost 48 hr) and have high levels of 53BP1 foci without PARPi treatment. These combined factors make it significantly more challenging to track the post-replicative foci specifically induced by PARPi over time and to analyze their 2D mobility ignoring movement along the z-axis.

Nevertheless, we performed this test and, in two independent experiments, observed a slight increase in DSBs mobility after myriocin treatment (see Figure below, for reviewers only). However, since the detected mobility is very low, compared to the one observed in MEFs (even below than in nocodazole-treated MEFs), we do not feel confident about including these results in the manuscript. We hope the reviewer understands our concern and agrees to leave the development of a consistent method to analyze DSBs mobility across different cell types for future studies.

3. Analysis of human cancer cell survival in Fig. 5: Only one PARPi concentration and a different graphical representation is used (bar diagram, linear y-axis). For better comparison and consistency, I suggest using 0.5 and 2 µg PARPi also in human cancer cell lines and display results in the same graphical representation as in the previous figures for survival curves.

Following the suggestion, we have now expanded the analysis to different concentrations of PARPi for both cell lines and their controls (new Fig. 5c,e,f and Extended Data Fig. 5c,d). The effect of PARPi is now clearer.

Minor points:

1. line 86, the term mobility is used twice: „The mobility of DSB mobility “

Thank you, fixed.

2. line 156, typo: “ Improtantly,...“

Fixed, thank you.

3. Figure 3e: graphs as mislabelled – the lowest curve (green) must be LMNA/C -/- and the black curve on top LMNA/C +/-

Fixed, thank you.

4. Extended Data Fig. 3i: Error bars are missing for the hLMNA+cre curve.

We apologize for the misunderstanding, but for the last point the error bar is very small and “hidden” by the symbol (Average: 1.05, SD=0.08).

5. Figure 3e: LMNA/C-deficiency already has an impact on cell survival upon BRCA-loss. Please show relative survival of the respective cell lines upon PARPi (as % to untreated) to allow a better comparison of the PARPi-effect on LMNA/C -/- cells to the LMNA/C +/+ counterparts.

In new Extended Data Fig. 2i,j and new Extended Data Fig. 4e, we show the relative survival upon PARPi, as requested by the reviewer. We would prefer to keep Fig. 3e and Fig. 4 in the main text as it is, because we believe that the reduced survival observed in BRCA1-null cells, in the absence of the PARPi, is a relevant feature of LMNA/C deletion.

6. Line 173: “Furthermore, we did not observe any reduction in 53BP1 levels in the absence of LMNA (Extended Data Fig. 3a)”. Extended Data Fig. 3a does not show a comparison of LMNA/C -/- and LMNA/C +/+ cells. Either this statement should be re-phrased or correct western blots need to be shown.

In the new Extended Data Fig. 3j, we have added the western blot showing unchanged 53BP1 levels in LMNA-null cells versus wt cells.

7. Figure 5a, please provide quantification

We have added the percentage of HCC-1937 cells with invagination with or without myriocin in a representative experiment in Fig. 5a.

Reviewer #2 (Remarks to the Author):

The manuscript by Faustini et al. investigates the involvement of microtubules, nuclear lamin proteins (Lamin A/C and Lamin B1), and sphingolipids in the repair of DNA Double Strand Breaks (DSBs), particularly in BRCA1-deficient cells treated with PARP inhibitors (PARPi). They demonstrate that increase in nuclear DSB mobility and envelope (NE) invaginations induced by microtubule dynamics and LaminA/C deficiency correlates with enhanced PARPi cytotoxicity in BRCA1-deficient cells. Additionally, depletion of SPTLC1, a subunit of the sphingolipid synthesis complex, increases NE invaginations and DSB mobility, similar to Lamin A/C depletion. Targeting sphingolipids with myriocin enhances PARPi efficacy in BRCA1-deficient cancer cells without affecting BRCA1-proficient cells. These findings suggest that NE remodeling, particularly through increased invaginations, could serve as a potential therapeutic strategy to enhance PARPi therapy efficacy in BRCA1-deficient cancers.

In general, this work is of high quality, the manuscript is well-written, addressing questions of great interest to researchers in the DNA repair and nuclear organization fields, with strong implication for cancer treatment.

We thank the reviewer for the time dedicated to our manuscript and their positive comments and support of our work.

This study nicely parallels a recently published study (Shokrollahi et al. NSMB) describing NE invaginations referred to as dsbNETs, which are likely identical to the NE invaginations observed in the present study (induced by microtubules and enriched in Lamin B1 in BRCA1-deficient cells treated with Olaparib). Affecting these dsbNETs (through KIF5B depletion) decreases sensitivity to PARPi and reduces the ability of PARPi to induce toxic misjoined chromosomes in BRCA1-deficient cells. Conversely, disruption of dsbNETs decreased BRCA1-deficient cells growth and tumorigenicity in mouse xenografts in absence of PARPi treatment. Do the author have an explanation for this opposite effect in presence or absence of PARPi?

The present study confirms that NE invaginations (dsbNETs) increase the PARPi sensitivity of BRCA1-deficient cells. Additionally it shows that increasing NE invaginations upon treatment with myriocin, a FDA approved drug, increases PARPi sensitivity of BRCA1-deficient cells, providing a proof of principle for cancer treatment.

Additionally, the authors show that taxol reduces both DSB dynamics and NE invaginations. However the authors did not test the effect of taxol on PARPi efficacy in BRCA1-deficient cells. The results of this experiment would be particularly relevant, given that taxol is one of the most important anticancer drugs, widely used in clinics for the treatment of ovarian and breast cancer. In this regard, authors should discuss published studies combining taxol (paclitaxel) and PARPi (olaparib) such as the work by Faschling et al. 2021 published in the Annals of Oncology and Zhang et al. 2022 in Am J Transl Res.

We thank the reviewer for this relevant suggestion. We tested different concentration of taxol in combination with PARPi and now show that, although taxol treatment is very toxic per se (new Extended Data Fig. 5e), in a sublethal concentration it partially suppresses the lethality of PARPi in uWB1.289 cells, while this is not observed in the same cells complemented with BRCA1 (new Extended Data Fig. 5g,h) confirming the requirement of dynamic microtubules for PARPi-sensitivity specifically in BRCA1-deficient cells. The same taxol concentration had no clear effect in HCC-1937 cells (new Extended Data Fig. 5f).

We have also added a comment in the Discussion section, where we refer to the two suggested papers: “On the contrary, the observed reduction in the PARPi efficacy after treatment with taxol in the BRCA1-deficient ovarian cancer uWB1.289 cell line would argue against the combined treatment of olaparib with paclitaxel, where olaparib has already been shown to not increase the efficacy of treatment with paclitaxel and carboplatinum (Faschling et al., 2021; Zhang and Zhang, 2022).”

Minor concerns:

- Some figures are missing statistics (extended data Fig1. b, Fig1. e difference between BRCA1F/F + PARPi -Taxol vs BRCA1F/F + PARPi +Taxol).

We have added the statistics to Fig. 1e and Extended Data Fig. 1b. There is no significant difference between +Cre+PARPi+taxol neither compared to noCrenoPARPi nor to +Cre+PARPi, indicating a partial suppression of the phenotype. As we added in the discussion “the reduction is only partial. While this could be simply due to the limited duration (1 hour) of taxol treatment, dictated by its toxicity, it is still possible that other cyto- or nucleoskeleton components could contribute to NE deformations, as nuclear actin or chromatin decompaction

- The conclusion on the top of p7 “These results indicate that the NE invaginations caused by the absence of Lamin A/C increase the mobility of one-ended DSBs, therefore increasing their misrepair” is too strong. Considering the pleiotropic effects of LMNA/C depletion (stabilization of 53BP1 or effect on DSB synapsis), NE invagination may not be causal. The authors should state that they observe a correlation.

We have rephrased the conclusions as suggested: “These results indicate that the NE invaginations, caused by the absence of Lamin A/C, correlate with higher mobility of one-ended DSBs and increased misrepair events, while neither of them appears to be affected by other kind of deformations such as NE blebs, as due to Lamin B1 absence.”

- The authors state that Lamin B1 organization is not perturbed upon shSPTLC1; however, they observed increased NE invaginations which is a phenotype typically observed upon LMNB1 overexpression. The authors should thus check LaminB1 level upon shSPTLC1 by western blots or quantification of IFs.

We performed this control by immunoblot, as suggested by the reviewer and found that lamin B1 levels are not altered in SPTLC1 depleted cells. This panel is included in (new Extended data Fig. 4a) in the revised manuscript.

- Extended data Fig. 1: Panel d is not well referenced in the figure legend.

Fixed (new Extended data Fig. 1f), thank you.

I would like to emphasize the clarity of this manuscript and the high quality of the data presented. Addressing the effect of taxol on PARPi efficacy in BRCA1-deficient cells will make this manuscript a perfect fit for publication in Nature Communications.

Thank you.

Reviewer #3 (Remarks to the Author):

In this manuscript, the authors suggest that nuclear deformations affect the mobility of DNA breaks. Increased DNA breaks' mobility is thought to increase sensitivity to the inhibition of PARP in cells that have lost BRCA1. The results, as presented, are hard to interpret and do not support the main conclusions of the manuscript. A major flaw with the experimental approach is that targeting the levels of the nuclear lamins or lipid synthesis will disrupt every aspect of cell physiology and have many pleiotropic effects. Both nuclear lamina and lipid synthesis are essential for viability. In addition, the authors overinterpret some of the results to draw hypotheses about the properties of the nuclear envelope. Without direct assays determining the properties of the nuclear membrane, it is unclear what the images of the nucleus obtained by microscopy mean. Although the data presented is exciting and, with the proper controls, the authors could establish a link between the nuclear envelope (NE) and DNA breaks, the main conclusions of the manuscript are not supported by the experimental approaches and results.

We thank the reviewer for the time dedicated to our manuscript and for their feedback. We have tried to address all the points raised by the reviewer and further clarified the text and we hope they now find the main conclusion supported by the data.

The manuscript, as presented, is hard to read and needs to be revised. It could be improved by addressing unclear terminology, overinterpreting the data, unclear labeling of figures, and spelling and grammatical mistakes.

The revised version of the manuscript contains several improvements, including new data, additional controls, corrected ambiguities and adjusted interpretation of some of the results as suggested by the reviewers.

Line 57: What are NE remodeling drugs?

To be more specific we have replaced it with “drugs increasing NE deformability”

In the introduction, the authors say that it is unknown whether microtubules (MT) promote the mobility of DSB in BRCA1-deficient cells (line 84). This statement contradicts the previous statement in the abstract, saying MT promotes the mobility of DNA DSBs (line 47).

Thank you for pointing out this incongruency. In the abstract we refer to our previous work showing the microtubule-dependent mobility of DSBs induced by ionizing radiation (IR), while in the introduction section, we focused on the mobility of DSBs generated by PARPi treatment of BRCA1 deficient cells. We intended to point out a structural difference between a proper (two-ended) DSB, generated by IR and one-ended DSBs, generated during replication of single-stranded breaks that accumulate in olaparib-treated BRCA1-deficient cells. This difference, along with the pathological relevance of one-ended DSBs, warranted further investigation of their mobility, especially considering that dynamic microtubules affect the repair of one-ended DSBs. In the revised version of the manuscript we have corrected the text, in order to make this distinction clear.

It is unclear what the viscoelastic properties of the NE mean (line 137).

We apologize for the mistake, we have rephrased it with: “viscoelastic properties of the nucleus”

How do viscoelastic properties promote and/or counteract MT forces? How can the properties of the NE be targeted? How can changing the properties of the NE affect the efficacy of drugs that inhibit PARP?

We assume that the reviewer is referring to this sentence in the introduction: “therefore, if the general viscoelastic properties of the nucleus could affect the microtubule forces promoting DSBs mobility and if the deformability of the nuclear envelope (NE) could be targeted to increase PARPi efficacy in BRCA1-deficient tumors is still unknown.”, where we expressed our hypothesis for the study. We have rephrase the sentence to make it more clear that they were our working hypothesis: “Therefore, if the general viscoelastic properties of the nucleus could support and/or counteract the microtubule forces promoting DSBs mobility and if they deformability of the nuclear envelope (NE) could be targeted to increase PARPi

efficacy in BRCA1-deficient tumors is still unknown.” We believe that our data showing that the mobility of DNA ends depends on the properties of the nuclear envelope, which we altered by deleting LMNA/C or depleting SPTLC1 (Fig. 2h and Fig. 4e), and correlates with increased lethality in cells that accumulate one-ended DSBs, as in the case of BRCA1-deficient cells treated with olaparib (Fig. 3e and Fig. 4i) support our initial hypothesis.

How does targeting of p53 immortalize MEF? This approach is a significant issue in the manuscript. Loss of p53 is not considered in the interpretations of the results. Also, MEFs without p53 tend to become tetraploid and may be genomically unstable.

Inactivation of p53 prevents the activation of senescence and/or apoptosis program in mouse embryo fibroblasts, in response to multiple stress stimuli (PMID: 2525423). Primary MEFs, with wild-type p53, stop proliferating within few passages and become senescent. To our knowledge, among the 3 main immortalization procedures used for mouse cells, direct p53 inactivation is the most mild and controlled one. Indeed, the other common strategies involve infection with the SV40-Large-T antigen, which is known to inactivate among other things both p53 and RB (PMID: 1652450) or spontaneous emergence of immortalized cells in extensive passages, as in the case of NIH 3T3 cells which likely generate a highly-diverse multiclonal population. It is indeed true that all immortalization strategies affect the biology of the cell lines on multiple layers, but this is an obligatory step for many experiments with cell lines. Proper controls, immortalised with the same strategy are important in this setting. In support of our choice of immortalization strategy, we would like to point out that p53 mutations occurs in more than 50% of cancers, including the ones we included in our study (new Extended Data Fig.5a).

In Fig. 1a, images of control cells are not shown: BRCA1 F/F – Cre with or without PARPi.

Thank you for the remark, we have added a new experiment with all the four conditions to show that treatment with PARPi triggers 53BP1 foci also in the presence of BRCA1, but to lesser extent than when BRCA1 is absent (new Extended Data Fig.1c,d)

Nuclear staining would be helpful in visualizing the nucleus in Fig. 1b.

We do not use a nuclear marker when we perform live-cell imaging because mCherry-53BP1 localize only inside the nucleus (DOI: 10.1038/nature07433; Fig.3c). However, we will make all the live cell imaging movies available, where the nuclear shape is more clearly visible.

What is the data that shows highly distorted nuclei?

In extended Fig. 1c, what are the discarded nuclei? What are the excluded nuclei?

We apologize for the lack of clarity. We have previously shown that large-scale nuclear deformations, rigid rotations, and expansion/contraction can affect the analysis of DSBs mobility by masking the local motion of the chromatin and introducing ambiguity in the analysis, so we developed a method to discard the nuclei undergoing such severe deformation and exclude them from the quantification of MSD or CD (Lottersberger et al., 2015; Faustini et al, 2024). We now explain these steps better in the main text of the revised manuscript: “We then analyzed the mobility of one-ended DSBs in BRCA1 deficient cells treated with PARPi as previously described (Lottersberger et al., 2015; Faustini et al., 2024). First, nuclei undergoing large-scale deformations that prevent unambiguous tracking of single foci mobility, were excluded from the analysis (Extended Data Fig. 1e). Then, we calculated the Mean Square Displacement (MSD) and the cumulative distance (CD) of all the traced 53BP1 foci (Fig. 1b,c and Extended Data Fig. 1f).” and in the legend of extended Data Fig. 1e: “Quantification of nuclei excluded from further analysis due to large-scale deformations as detected by tracing the mobility of the mCherry-53BP1 foci 72 h after BRCA1 deletion with Hit&Run Cre and 6 h after treatment with PARPi and DMSO, taxol (1 h) or nocodazole (2 h).”

There is no figure legend for extended Fig. 1d.

Fixed it, thank you (now Extended Data Fig.1f)

The cumulative distance differences are minimal. Most data points for all three conditions lie between 1 and 3 μm .

We agree with the reviewer that the CD is less informative than MSD, this is why we added it to the extended data. However, we think this is still a relevant information, the data are consistent with previous analysis and the significance was calculated as described and reported. We have explained it better in the text: “As expected, DSBs induced by PARPi in the absence of BRCA1 roam the nucleus similarly to dysfunctional telomeres and IR-induced DSBs (Lottersberger et al., 2015; Dimitrova et al, 2008), with a final MSD of 0.3 μm^2 and a median CD of 2.4 μm after 10 min (Fig. 1b,c and Extended Data Fig. 1f). Furthermore both taxol, a microtubule stabilizer, and nocodazole, a microtubule depolymerizer, reduced the mobility of DSBs, without affecting the number of excluded nuclei (Fig. 1b,c and Extended Data Fig. 1e,f)”.

The effects on nuclear morphology seem minimal in Figure 1d. Also, the analyses of the nuclear morphology of -Cre +PARPi, +taxol alone, or no Cre +PARPi + taxol are not shown.

We have added the analysis performed by IF in new Extended Data Fig. 1h,i showing that increased deformations are observed only in PARPi-treated BRCA1-deficient cells, and that taxol does not affect the normal level of invaginations. The ultra-fine, low-throughput analysis by EM was applied only to the most relevant conditions no Cre no PARPi, + Cre + PARPi and+ Cre + PARPi + Taxol.

What is the evidence that MT cause nuclear invaginations?

In addition to the EM analysis, in the revised version of the manuscript, we show that treatment with taxol for 1 h at the end of PARPi treatment partially reduced the number of invaginations, indicating that dynamic microtubules contribute to their formation. We have expanded the discussion on why the reduction is not complete: “However, the reduction is only partial. While this could be simply due to the limited duration (1 hour) of taxol treatment, dictated by its toxicity, it is still possible that other cyto- or nucleoskeleton components could contribute to NE deformations, as nuclear actin or chromatin decompaction”

The data in Fig 1e shows a change from 1 to 2 nuclear abnormalities per cell. Is this change physiologically relevant?

The increase that we can measure in this experimental setting is only two-folds, but it is significant. Furthermore, considering that after 6 h PARPi only about 50% of cells are responding, our measurements represent an underestimation of the total level of deformations that these cells experience over time. This is further exacerbated by the fact that we are looking only at one z-stack in a fixed samples, and these deformations could be highly dynamic.

It is unclear how, if, or what does it mean MT directly poking the nucleus. What are the data showing MT poking NE?

We have rephrased the indicated statement with “dynamically deforming the nucleus”, which provides a better description of the images.

The signal of lamin B1 in Ext Fig. 1e does not represent NE invaginations. Are these images from confocal microscopy?

Staining with Lamin B1 antibodies is an established method to detect deformations and invaginations in the nucleus. All images were all aquired with widefield DeltaVision Microscope, with 16-21 stacks every 0.3 μm , as shown below. Invaginations were visualized/quantified focusing on the middle stacks, and we have now added an example of the top/middle/bottom stacks in in new Extended Data Fig.1g.

It is unclear what is presented in Ext. Fig 1f. What does "-" mean? Normal?

We have rephrased the legend: "Quantification of nuclei with blebs (blebs), invagination (inv) or no deformations (-)".

Control experiments are missing. +Cre no PARPi or PARPi alone.

We have added these two controls, which show an intermediate phenotype, not significantly different from untreated cells (new Extended Data Fig. 1h).

Figure 1d: how can you tell a vertical invagination from electron microscopy? What is the separation of nuclear areas? The middle panel shows a micronucleus. What data shows that taxol reduces the separation of nuclear areas?

We apologize for not having better described the structures we scored as deformations. We have now added a better description in the text: "EM sections showed also the appearance inside the nucleus of distinct nuclear areas surrounded by an electron-dense material, compatible with transversal sections of NE invaginations, and distinct nuclear regions that appear separated by a very narrow cytoplasmic channel, which are compatible with longitudinal sections of NE invaginations" and "Furthermore, both IF and EM analysis showed a partial reduction in the number of NE invagination after a 1-hr treatment with taxol (Fig. 1d,e and Extended Data Fig. 1i), indicating that, in the presence of one-ended DNA breaks, microtubules dynamics contributes to promote NE invaginations that may then mediate DSBs mobility and mis-repair".

The data presented in Figure 1 does not strongly support the conclusion that microtubule dynamics promote NE invaginations and mediate DSB mobility and misrepair.

Preventing microtubule depolymerization with taxol clearly reduces both nuclear envelope invaginations and DSB mobility. Furthermore, this result is consistent with the effect on mobility and repair, shown in a previous study by us (Lottersberger et al 2015). In addition, the data in Fig. 1 are further supported by the observation that taxol treatment reduces mobility in different genetic contexts (Fig. 2 and Extended Data Fig. 2) or the lethality of PARPi (new Extended Data Fig. 5). Based on these considerations, we believe that our conclusion provides a fair interpretation of the results.

In Figure 2a, it is unclear how NE invaginations are defined.

We have improved the contrast to make the invaginations more visible.

Experiments after deleting lamin A/C are hard to interpret since lamin A/C is an essential gene. Loss of nuclear lamina will affect every aspect of cell physiology, including transcription, nuclear pores, cell division, and more.

Lamin A/C is not essential in mice, homozygous mutant pups are viable (doi: 10.1083/jcb.147.5.913), and MEFs survive similarly with and without LMNA/C (Fig3d and new Extended Data Fig. 2a). However, we completely agree that many other aspects of cell physiology could be affected by LaminA/C deletion and this is why we explored the role of nuclear deformability in DSBs repair by altering the shingolipids synthesis. We have added this remark at the beginning of the third results chapter: “Lamin A/C affects several aspect of cell physiology, including chromatin organization and gene regulation (de Leeuw et al., 2018)”.

Taxol reduces MSD with or without LMNA, which suggests that MT's effect is independent of LMNA. The data presented in this manuscript do not support the conclusion that Lamin A counteracts the forces generated on the NE by MT.

The observation that the high mobility observed in LMNA-deficient cells is strongly reduced by taxol treatment indicates that dynamic microtubules are responsible for such mobility. However, since the reduction is not complete, it is possible that other factors, including the other cytoskeleton components, play also a role. Since Lamin A/C form a mesh that confers NE rigidity, it is reasonable to suggest that this mesh may counteract the MT dynamics. We have clarified this part better in the text and in the conclusions: “. This increase in DSB hyper mobility in the absence of Lamin A was strongly reduced by taxol treatment (Fig. 2g,h and Extended Data Fig. 2d-f), indicating that dynamic microtubules are the main cytoskeleton force responsible for the mobility and suggesting that the Lamin A network, and not Lamin B1, counteracts the microtubule forces generated on the NE by increasing NE rigidity.”.

In Figure 3a, do the arrowheads point to translocations? How can one tell what a misrejoined chromosome is from these images?

We apologize for having misplaced some arrows. We have now moved them so that they all point to the centromeres (identified by their telocentric brightness) of chromosomes involved in mis-rejoining events, either due to one chromosome arm wrongly attached to another chromosome arm or to a second centromere. We added it in the legend as “Arrowheads indicate centromeres of chromosomes involved in mis-rejoining events”.

Figure 3e shows that PARPi kills 99% of cells with or without lamin A/C. Also, Cre alone reduces viability by 90%, indicating that BRCA1 deletion makes cells very sick.

Indeed deletion of BRCA1 reduces the viability (of otherwise wt cells) of 90%, with PARPi increasing the lethality by another 90%. Both viability and PARPi hypersensitivity are restored by deletion of the 53BP1-axis. We have rephrased the text and added this information in the introduction section “Ablation of any of these factors restores resection and, therefore, HR in the absence of BRCA1, thus promoting survival and increasing resistance to PARP inhibition.”.

Data in Ext Fig 3g shows that re-expression of LMNA minimally suppresses % of mis-rejoined chromosomes.

The percentage of mis-rejoined chromosomes after PARPi treatment drops by about 50%. We consider the suppression substantial.

It is unclear how sphingolipids reduce the fluidity and the curvature of cell membranes.

Sphingolipids have high bending rigidity. We have added this information in the text.

SPTLC1 is an essential gene, and lipid synthesis is essential for viability (<https://depmap.org/portal/gene/SPTLC1?tab=overview>).

SPTLC1 is considered an essential gene, however, we did not delete it, just deplete it with shRNA. We have added now a new Extended Data Fig. 4b to show that such depletion does not significantly reduce cell viability.

The results presented in Figure 5 are expected. The combination of PARPi and myriocin is more toxic than either condition alone. The analysis of a few cell lines and one drug concentration is insufficient to draw general conclusions about cancer and SPT or PARP inhibition.

In the new Extended Data Fig. 5b of the revised manuscript, we now show that the effect of myriocin treatment alone on cellular viability is minimal, too small to explain the drop in viability in the combined treatment. Furthermore, we have extended the analysis covering additional concentrations of PARPi and we have normalized the results on myriocin treatment. These experiments show an increased sensitivity to PARPi due to combined treatment with myriocin in the most commonly used cell lines lacking BRCA1 but not in cells expressing BRCA1, or in the MCF-7 cell line, which is the most sensitive to myriocin (new Fig. 5c,e,f and new Extended Data Fig. 5c,d). Furthermore, our new analysis with taxol + PARPi shows that, in this case, a drug which is toxic on its own, does not increase the toxicity of PARPi further, but, on the contrary, it leads to partial suppression of its lethality, likely through reduction of DSB mobility (new Extended Data Fig. 5e-h).

Reviewer #1 (Remarks to the Author)

Faustini and co-workers have added substantial amount of work to improve the manuscript and to address reviewers concerns.

I appreciate their effort and the improved quality of the presented work.

After addressing the two following minor points, I see this work ready for publication in Nature Communications:

We thank the reviewer for the positive revision.

1. In line 121 of the revised manuscript, authors state:

“The fraction of 53BP1-positive cells, however, was reduced from 50% to 35% compared to the one observed in BRCA1-deleted cells (Fig. 1a and Extended Data Fig. 1c,d).”

In the data provided in the respective figures, no percentages are given. The graph in Extended Data Fig 1d shows the “number of 53BP1 foci/nucleus”.

Please re-phrase or provide missing data.

We apologize for the confusion, and we rephrased indicating the median foci/cells, as shown in the figure:

“However, the median number of 53BP1 foci per cell, compared to BRCA1-deleted cells treated with PARPi, was reduced from about 20 to 1”.

2. I asked the authors to correct labeling of their survival curves in Figure 3a (minor point 3).

Surprisingly, I still find Figure 3a mis-labeled in the revised manuscript. Please change accordingly.

We apologise for this reiterated problem, we made a mistake in saving the figure. It is now corrected.

Reviewer #2 (Remarks to the Author)

The authors have addressed all my concerns in the revised manuscript

We thank the reviewer for the positive revision.

Reviewer #3 (Remarks to the Author)

The revised manuscript clarifies many of the confusing points I raised before and is now easier to understand.

However, several issues remain to be addressed, and the data provides correlative results to support the main conclusions of the manuscript.

We thank the reviewer for the positive comments. We have addressed the remaining issues in the revised manuscript.

Almost every western blot (WB) in the manuscript is missing loading controls and molecular markers:

We apologize for this. We have always controlled for equal loading by the ponceau staining and by re-hybridizing the membranes or by running a parallel WB for actin or Lamin B1. We have added all these controls in the revised version of the manuscript along with the annotation of the approximate molecular markers.

WB of Figure 1a is missing loading control. Levels of Chk2 are variable.

We thank the reviewer for noticing that the western blot for Chk2 has a dot at one edge of the first well which could be misleading. We have run another western blot, with actin as loading control, which shows that the Chk2 levels do not change. We have included this blot in the revised Fig.1a.

WB in Figure 2b and 2c are missing loading controls. Lamin A/C should be 2 bands.

We have added the ponceau S staining for Fig.2b and actin for Fig.2c as loading controls.

As for the only one band in the Lamin A/C WB, the reviewer is correct. At the time we performed that experiment we used an antibody from Santa Cruz that recognizes only Lamin A (sc-518013). We apologize for the mislabeling and are grateful to the reviewer for pointing this out, allowing us to amend it. We have corrected the labels in the Fig. 2b and the corresponding legend.

WB in Ext. Fig. 3a is missing loading controls. The bands are for LMNA/C or LMNA/C-GFP? No MW markers are shown.

We have added as loading control Lamin B1 and the MW markers. As for the Lamin A band, the reviewer is correct and we have now corrected with GFP-hLMNA. Thank you for pointing this out.

WB in Ext. Fig 3j, no loading control.

We have moved the 53BP1 western blot shown before in Extended Data Fig.3j to the revised Fig.2b, adding ponceau S staining as loading control.

Figure 4a, WB shows loading control Lamin A/C. There are 2 bands as expected. These two bands are not seen in the manuscript in other westerns of Lamin A/C.

As mentioned above, in the other blots, where only one band is visible, we have used an antibody from Santa Cruz that recognizes only Lamin A. We have corrected the figure labels and the figure legends accordingly.

Regarding the loading control, we have used Lamin A/C since its levels do not change in the absence of SPTLC1. As a further control, the Ponceau S staining was confirms equal loading (Figure 1 for Reviewers):

Figure 1 for Reviewer: Ponceau S staining for WB in Fig.4a

Furthermore, we have re-run the same samples using actin as loading control (Figure 2 for Reviewers):

Figure 2 for Reviewer: immunoblot of deletion of BRCA1 and depletion of SPTLC1 in BRCA1^{F/F} MEFs untreated or treated with Hit & Run Cre. Actin is shown as loading control.

In line 121, 53BP1 positive cells was reduced from 50% to 35% compared to the one observed in BRAC1-deleted cells. Neither Fig. 1a nor Extended Fig. 1c, d show these numbers.

This point was raised also by reviewer 1, we apologize for the confusion. We previously provided the fraction of cells with more than 20 mCherry-53Bp1 foci as an arbitrary threshold of DNA damage activation, but we now rephrased the text indicating the median number of foci per cell, as shown in the figure:

“However, the median number of mCherry-53BP1 foci per cell, compared to BRCA1-deleted cells treated with PARPi, was reduced from about 20 to 1”

The effects of taxol or nocodazole in DSB mobility in WT +PARPi are not shown.

Indeed, here we did not investigate the mobility of DSBs induced by PARPi in recombination-proficient cells. One reason is that in BRCA1-proficient cells, the number (and the brightness) of the mCherry-53BP1 foci formed after PARPi treatment is much lower than in BRCA1-deficient cells (Extended Data Fig. 1c,d). This would make it very difficult to quantify foci mobility with the same settings used for BRCA-deficient cells and it would require carefully adjusting the experimental conditions for accurate measurement of the mobility of fewer and less bright foci. Furthermore, while we agree that it would be of high interest to investigate the effect of DSB mobility in the repair by homologous recombination (as it would happen in BRCA1-proficient cells), we believe that this is beyond the scope of the current study, which instead focus on the the effect of mobility on the miss-repair (by NHEJ) of the one-ended DSBs that accumulate in BRCA1-deficient cells.

In Ext Fig. 1h, PARPi treatment did not cause a significant increase in NE invaginations in BRAC1+ cells or BRAC1- cells. The comparisons show no significance. There is a significant effect on BRAC1+ /PARPi and BRAC1- /PARPi (red bars); these results suggest that BRAC1 deletion in the context of PARPi causes changes in the NE. PARPi treatment alone does not cause NE invaginations. These results are confusing as to what exactly is causing NE defects.

We apologize for the confusion. Our intention was not to imply that PARPi treatment alone triggers NE invaginations. Rather, we aimed to convey that the extent of NE deformations correlates with the degree of the activation of DNA damage response. In fact, BRCA1 deletion or PARP inhibition alone trigger only a mild activation of the DNA damage response, compared to the combination of BRCA deletion and PARP inhibition (as shown in Fig. 1a for Chk2 phosphorylation and Extended Data Fig. 1c,d for the mCherry-53BP1 foci accumulation). Accordingly, each condition alone seems to increase NE invaginations, although not to a level that reaches statistical significance in our experimental setting. On the contrary, the higher level of DDR activation in BRCA1-deficient cells treated with PARPi is associated with a significant increase in NE invaginations.

Consistently, as the reviewer points out, NE invaginations increase more (although not significantly) after BRCA1 deletion alone, than after PARPi treatment alone. Indeed, in BRCA1-deleted cells, the main DSB response kinase, ATM, is also more active compared to BRCA1-wt cells treated with the PARPi, as shown by the slightly higher levels of Chk2 phosphorylation (Fig. 1a), supporting the view that higher levels of DNA damage response cause higher levels of NE invaginations.

This view is further supported by an independent study that was published while this paper was in revision (Shokrollahi et al., NSMB, 2024; PMID: 38632359; Figure 3 for Reviewers only) showing that NE deformations (here called nuclear envelope tubules) are formed after massive induction of DSBs with etoposide, and that these

deformations are dependent on the DNA damage response kinases ATM, ATR and DNA-PK.

[REDACTED]

Figure 3 for Reviewer only (from Shokrollahi et al., NSMB, 2024)

We added the clarifications in the text for both the DNA damage response and the invaginations:

“Three days after BRCA1 deletion, we observed some phosphorylation of the DNA damage kinase Chk2 and no accumulation of mCherry-53BP1-2 foci, while treatment with the PARPi olaparib for 6 hours caused robust phosphorylation of Chk2 and the formation of numerous (> 20) clear mCherry-53BP1-2 foci in about half of the cells (Fig. 1a and Extended Data Fig. 1a-d). Importantly, 6 hours of PARPi treatment in BRCA1-wild type cells, where DSBs should be repaired by HR, were not enough to trigger Chk2 phosphorylation, although they were enough to induce the formation of few mCherry-53BP1 foci. However, these foci are less bright and less common compared to BRCA1-deleted cells treated with PARPi (Fig. 1a and Extended Data Fig. 1c,d; median number of foci/cell: 1 versus 20), indicating that the activation of DNA damage response is weaker and/or that the DSBs are repaired more quickly.”

“Indeed, after deletion of BRCA1 and treatment with PARPi for 6 hours, we detected a significant increase in NE invaginations compared to the relatively smooth NE of BRCA1-proficient untreated cells, while no formation of blebs was observed (Extended Data Fig. 1h). On the contrary, the treatment with PARPi in BRCA1-proficient cells or the deletion of BRCA1 without PARPi treatment caused only a slight, and not statistically-significant, increase in NE invaginations, when compared to the control cells (Extended Data Fig. 1g,h). Since both PARP inhibition or BRCA1 deletion alone also cause only a mild activation of the DNA damage response (Fig. 1a and Extended Data Fig. 1c-d), these data indicate that the NE shape is altered in response to DNA damage response activation, consistently with a recent study showing that the DNA damage kinases ATM, ATR and DNA-PK contribute to the formation of Lamin B-rich tubules in human cells after DSBs induction with etoposide”

Which target of the DNA damage response kinases causes the NE deformations is still unclear. We discuss the different hypothesis currently formulated, mostly by others, as the microtubules (Ma et al., JCB, 2021), the LINC complex component SUN1 (Shokrollahi et al., NSMB, 2023), Lamin A/C (Kovacs et al., Mol Cell, 2023; Joo et al., Mol

Cell, 2023), nuclear actin (Lamm, N. et al. NCB, 2020) or chromatin decompaction (Dos Santos al., NAR, 2021), but we believe that further studies beyond this current work are required to completely elucidate this complex phenomenon.

The EM images show minimal changes in the nuclear envelope morphology. The middle image shows a micronucleus. The quantification in Fig 1e shows that PARPi and BRCA1 deletion increases the nuclear abnormalities from 1 to 2 per cell. It is unclear whether this effect is physiologically relevant and whether or how 1 more nuclear invagination per nucleus could account for changes in DSB mobility in the whole nucleus.

Indeed, by EM we can measure a median increase of NE deformations from 1/cell in untreated BRCA1-proficient cells to 2/cell in BRCA1-deficient PARP-inhibited cells, and we believe this to be physiologically relevant. In fact, another way to visualize this set of data is that the fraction of nuclei with no deformations decreases from 45% to 10%. Furthermore, we anticipate that the EM analysis would underestimate the number of events occurring in the whole nucleus of cells experiencing high levels of DNA Damage Respose activation considering that:

- only about 50% of BRCA1-deficient cells are undergoing high levels of DNA Damage Respose activation in the presence of PARPi (as seen by the formation of mCherry-53BP1-2 foci in Extended data Fig.1a-d), and we cannot select for them in EM
- the EM images provide a single Z-stack view
- such deformations could be highly dynamic and therefore are more difficult to estimate in a single snapshot

Indeed, preliminary data obtained by live-cell imaging of MEFs overexpressing GFP-hLMNA show how nuclei can undergo dramatic deformations over just 10 min (Figure 4 for Reviewers only). As the overexpression of hLMNA could affect NE structure (for example inducing blebs as shown by Piekarowicz et al., Chromosoma. 2017; PMID: 27534416), we do not feel confident about including these results in the manuscript and we hope the reviewer will agree to leave this analysis for future studies with better suited live-cell markers.

Figure 4 for Reviewers only: montage of the projection of live-cell imaging of a BRCA-deficient cell treated with PARPi and overexpressing GFP-hLMNA. The images were acquired every 30 sec for 10 min total.

As for the structure observed in the middle image of Fig.1d, we agree that it could be consistent also with a micronucleus structure. However, since it is quite large and it is separated from the main nucleus by a very narrow cytoplasmic channel, we believe it is also compatible with a region of the nucleus separated from a very long longitudinal, transient invagination. Given that this type of image analysis is subjective in nature, the importance of providing a blind quantification, and the observation that also this kind of events are not induced in BRCA1-deficient cells treated with PARPi and taxol, we think it is essential to include them in the scoring of NE deformations. However, we changed the text to include the uncertainty of the nature of these structures:

“... distinct nuclear regions that appear separated by a very narrow cytoplasmic channel, which could be compatible with longitudinal sections of NE invaginations”

Line 149-150: The taxol effect in Ext Figure 1i shows no significance, contradicting the text.

We apologize for the lack of clarity in the text, however, we don't think the data in Extended Data Fig.1i contradict our conclusions in the text. In the presence of taxol, we do not observe a statistically significant increase in NE invagination from the control cells (BRCA1-positive no PARPi). In other words, when compared to the wt condition, the presence of taxol clearly prevents the significant increase in NE invaginations observed after DNA Damage Response activation.

This result indicates that dynamic microtubules participate to the formation of NE invagination when there are DSBs.

This said, we agree that taxol treatment causes only a partial reduction of NE invaginations rather than complete elimination. This intermediate phenotype could be due to the short treatment with taxol (1 h) compared to the long treatment with PARPi (6 h). However, it is also possible that other cyto- or nucleoskeleton components could contribute to promote NE deformations, together with dynamic microtubules.

To address the reviewer's comment and represent more clearly and unambiguously the results in Ext fig 1i, we have corrected the text as follows:

“Furthermore, both IF and EM analysis showed that the addition of taxol for one hour prevents the significant accumulation of NE deformations observed in BRCA1-deficient cells treated with PARPi (Fig.1d,e and Extended Data Fig. 1i), indicating that, in the presence of one-ended DNA breaks, microtubule dynamics plays a major role in the formation of NE invaginations. However, taxol treatment reduced only partially (and not statistically-significantly) the number of NE abnormalities in BRCA1-deleted cells treated with PARPi, suggesting that other cytoskeletal or nucleoskeletal components may contribute to NE deformations alongside dynamic microtubules.”

As a side note, in Shokrollahi et al., NSMB, 2024 (PMID: 38632359; Figure 3 for Reviewer only), the NE deformations induced by etoposide are completely abolished by

nocodazole treatment, supporting our conclusion, although the results with the different drugs inducing DSBs and affecting microtubules is different, as we explain in the discussion.

Nocodazole does not increase NE invaginations in PARPi/BRCA1- cells (Ext Fig 1j last two bars) but reduces mobility (Ext fig 1f). These data contradict the main point of the manuscript that NE inv causes DNA mobility.

Line 155: “NE invaginations is not enough to promote mobility”. The paper is confusing because it is unclear what the relationship between deformability and nuclear envelope invaginations and nuclear blebs is.

This is a very important point, and we apologize for having not being able to present it clearly. It was not our intention to imply that NE invaginations alone are sufficient to promote DSB mobility. On the contrary, we conclude that the mobility of DSBs has a strict requirement for the dynamic contraction/expansion of microtubules, even in the presence of NE invaginations. This conclusion is also in agreement with our previous work on microtubules promoting the mobility of unprotected telomeres and IR-induced DSBs (Lottersberger et al., Cell, 2015; PMID: 26544937; Figure 5 for Reviewers only).

[REDACTED]

Figure 5 for Reviewer only: microtubules promotes the mobility of dysfunctional telomeres (induced by TRF2 deletion) and IR-induced DSBs (from Lottersberger et al., Cell, 2015)

However, we would like to point out that – so far – we have never observed high DSB mobility in the absence of NE invaginations. Instead, a significant increase in NE invaginations (as in LaminA/C-deficient cells or in SPTLC-depleted cells) correlates with a significant increase in DSBs mobility and mis-repair. Therefore, we propose that NE deformability affects the mobility of DSBs and therefore their repair in BRCA1-deficient cells, but only in the context where dynamic microtubules can promote mobility. In other words, in a more deformable nucleus, dynamic microtubule increase the mobility and therefore the rate of mis-repair by NHEJ of one-ended DSBs.

In the revised version of the manuscript we state that NE invaginations per se are not enough to promote mobility:

- Results: “the formation of NE invaginations per se is not enough to promote DSBs mobility”;
- Discussion: “Importantly, dynamic microtubules are still essential to promote DSBs mobility even in the presence of NE invaginations.”

In order to avoid further ambiguity in this important point, we have now modified the title and the first line of the Discussion, emphasizing the concept of NE deformability instead of focusing on its manifestation as NE invaginations:

-Title: “Nuclear deformability increases PARPi sensitivity in BRCA1-deficient cells by increasing microtubule-dependent DNA break mobility”

-Discussion: “Here we show that an increase in NE deformability promotes the microtubule-dependent mobility and mis-joining of one-ended DSBs when HR is compromised. Such deformability has to take the form of invaginations and can be induced by microtubules in the physiological context of the cellular response to DSBs, as shown also by others 25,37,38, or by artificially affecting the nuclear Lamin A/C network or the sphingolipids composition of the NE as shown here”

Figure 2c points to an effect of BRCA1 levels upon knockdown on lamin B.

We agree with the reviewer that there is a slight difference in BRCA1 levels with the second shRNA against Lamin B1, as clear also in the new western blots provided. Since this shRNA is the less efficient, we did not explore this phenotype further.

Although the reviewer is not suggesting this, we would still like to clarify that this observation does not affect in any way the conclusions of this experiment, which compares the effect of LaminB1 on DSB mobility and miss-repair after BRCA1 deletion and not when BRCA1 is active.

The MSD of LMNA^{-/-} treated with taxol is greater than control cells (Figure 2h). This indicates that microtubules (MT) are not the major force for DSB mobility. Because taxol reduces DSB mobility by a similar extent (50-60% of the untreated) with or without Lamin A the data in Figure 2h argues that DSB dependence on MT is independent of Lamin A.

As the reviewer points out, taxol treatment reduces DSB mobility by about 50% in the presence or absence of LaminA/C. Considering also a certain degree of noise in the experiments that measure DSB mobility, these results clearly indicate that microtubules are a major force for DSB mobility.

To support this conclusion, we have performed the analysis of DSBs mobility in taxol-treated SPTLC1 depleted cells showing that in this case the higher mobility is completely abolished by taxol (Figure 6 for Reviewers only). If the reviewer thinks that this analysis would strengthen the conclusions, we can include it in the manuscript.

Figure 6 for Reviewer only: taxol abolish the high DSBs mobility due to SPTLC1 knockdown. (a) discarded cells, (b) cumulative distance, (c) MSD of mCherry-53Bp1-2 foci in BRCA1-deleted MEFs treated with PARPi for 6 hours after deletion of SPTLC1 with shRNA and/or addition of taxol for 1 h before imaging. Data are from 3 independent experiments.

However, we agree with the reviewer that taxol does not reduce completely DSBs mobility in the absence of Lamin A/C as it does in the LMNA/C WT cells (or SPTLC-depleted cells), suggesting that LMNA deletion might increase the sensitivity of the NE to other cytoskeleton components and, therefore, that other mechanical forces play a role in DSB mobility in the absence of Lamin A/C. We have included this possibility in the text:

“Furthermore, LMNA deletion, but not Lamin B1 depletion, caused higher mobility of the DSBs induced by PARPi in BRCA1-deficient cells (Fig. 2g-i, and Extended Data Fig. 2d-h), similar to previous data obtained after DSBs induction with etoposide 29,30, indicating that the Lamin A network, but not Lamin B1, prevents DSBs mobility by increasing NE rigidity. This increase in DSB hyper mobility in the absence of Lamin A was strongly, although not completely, reduced by taxol treatment (Fig. 2g,h and Extended Data Fig. 2d-f), indicating that dynamic microtubules are the main, but perhaps not the only, cytoskeleton force responsible for DSBs mobility in the absence of Lamin A/C.”

The comparison of the data in Ext Figure 2f vs. 2h shows control cells cumulative distance of 2.367 (2f) and 2.531 (2h). That’s a difference of 0.164 between experiments. The difference between control and taxol is less than that ($2.367 - 2.254 = 0.113$), indicating that the effects of taxol are within the error of biological replicates.

We agree with the reviewer that the effect of taxol on DSBs mobility is not so evident when we look at cumulative distance (CD). However, it is consistent and significant in the 3 independent replicates shown in Extended Data Fig.1f (no taxol: 2.380; + taxol: 2.098), the 3 independent replicates shown in Extended Data Fig.2f (no taxol: 2.367; + taxol: 2.254), and the 3 independent replicates shown above in Figure 6.b for Reviewers (no taxol: 2.321; + taxol: 2.194).

Importantly, we would like to emphasize that all our main conclusions are based on the results obtained by measuring the Mean Square Displacement (MSD), which is considered a more sensitive metrics for quantifying particle mobility than CD. This is because MSD differentiates between a consistent movement from the initial position and an unproductive back and forth movement. The former is anticipated to be more biologically relevant in promoting the mis-joining of distant DNA ends. Indeed, the effect of taxol on DSBs is much more evident in the MSD, consistently reducing the final MSD at time point 10 min of 50% (as previously discussed) from about 0.3 to about 0.15 μm^2 (Fig.1c; Fig.2h; Figure 6.c for Reviewers), leading to the conclusion that dynamic microtubules play a major role in DSBs mobility.

The variability pointed out by the reviewer is expected between experiments that are not performed at the same time, due for example to the title of the virus used to delete BRCA1. In particular, we would like to point out that the cells used as controls in

Extended data Fig. 2f and 2h do not represent biological replicates since they were treated differently: in Extended data Fig. 2f (as in Extended data Fig. 1f), they are the original BRCA1F/F MEFs while in Extended data Fig. 2h, they are BRCA1F/F MEFs infected with the retroviral empty vector control pSuperior and then grown in puromycin for a couple of days before Cre-mediated deletion of BRCA1.

Data in Ext Figure 2e show that about 80% of nuclei with LMNA^{-/-} are excluded. This suggests that the 20% of cells analyzed may not represent the major physiological consequences of LMNA depletion and that the differences in DSB mobility are minor phenotypes in these cells.

As we have described previously (Lottersberger et al., Cell, 2015; PMID: 26544937; Figure 7 for Reviewers only), we systematically remove from foci analysis all the nuclei that are undergoing severe deformations in all conditions. This is because we noticed that NE deformations are quite frequent in MEFs and prevent the correct quantification of foci movement inside the nucleus. Therefore, we had to take this step in order to be more rigorous in our conclusions. However, this step does not lead to overestimation of the mobility, neither does it report a phenotype that is true only for a minority of cells. On the contrary, by excluding deformed nuclei we probably underestimate the differences in DSB mobility. In Figure 8 for Reviewers only, we report the MSD, without excluding the the distorted nuclei for one of the three experiments used for Fig.2h. In this case the MSD values are higher for all samples, but more so for the LMNA/C null MEFs.

[REDACTED]

Figure 7 for Reviewer only: identification of the distorted nuclei (from Lottersberger et al., cell, 2015).

Figure 8 for Reviewer only: MSD of mCherry-53BP1-2 foci in LMNA/C+/+ or LMNA/C-/- MEFs after deletion of BRCA1 and treatment with PARPi with/without taxol in 1 representative experiment 8 from Fig. 2g-h and Extended Data Fig. 2d-f). The foci MSD analysis was conducted before (left) or after (right) removal of the highly distorted nuclei. The fraction of the removed distorted nuclei is: LMNA+/+: 46 %, LMNA+/+ + taxol: 44 %, LMNA-/-: 83 % and LMNA-/- + taxol: 50%.

Line 187 states that Lamin B depletion does not affect viability. Figure 3f shows that shLB1 + Cre has significantly lower viability than -Cre.

The decrease in viability in the + Cre cells is due to the deletion in BRCA1, which is well known to affect cell viability, not to the depletion of Lamin B.

Our comment (in line 187 of the previous version of the manuscript) was referring to the effect of Lamin B1 depletion after the removal of BRCA1, where both shRNAs against Lamin B1 do not decrease viability compared to the empty vector control (or LMNA/C deletion):

“Importantly, after removal of BRCA1, LMNA deletion, but not Lamin B1 depletion, further reduced cell survival, exacerbated the hypersensitivity to PARPi, and induced a significant increase in the formation of aberrant chromosomes (radials) after PARPi treatment.”

If the reviewer were instead referring to the effect of Lamin B1 depletion on the PARPi sensitivity of the HR proficient MEFs (the three lines at the top of the graph), it may be true that there is a slight decrease. While this result might support the view that the response to DSBs in BRCA1 positive and negative cells is differently regulated by mobility, the effect is really minimal, and goes beyond the main focus of the study, which is the role of mobility in HR(BRCA1)-deficient cells. We just added a comment in the text for sake of clarity:

“Consistent with a previous report showing that LMNA-deficient MEFs are not hypersensitive to etoposide 30, Lamin A/C deficiency did not affect the survival to PARPi of BRCA1-proficient MEFs, which can repair DSBs by HR (Fig. 3e), while Lamin B1 depletion slightly reduced it (Fig. 3f).”

The results describing the effects of sphingolipid inhibition on DSB remain correlative observations and require further investigation. Without assessing the effects of sphingolipid inhibition on the biophysical properties of the nuclear membrane, these results remain speculative.

While we did not check the effect of sphingolipids reduction on the biophysical properties of the membrane, we confirmed their role in preserving NE shape, which is consistent with the sphingolipids' known role in promoting “*The transition from a thin and loosely packed membrane into a thick and rigid one*” (from Joost and Menon, Nature, 2014; PMID: 24899304) and the observation reported by Hwang et al, Cell Reports, 2019 (PMID: 31747614) of a significant increase in NE deformation in the human epithelial cell line RPE and primary human skin fibroblasts (HSF) after inhibition of SPTLC with myriocin or depletion of SPTLC1/2 with shRNAs (Figure 9 for Reviewers only).

[REDACTED]

Figure 9 for Reviewers only: SPTLC inhibition promotes NE deformations in human cell lines RPE-1 and HSF (from Hwang et al, Cell Reports, 2019)

In fact, we show that shSPTLC1 induces NE deformation in MEFs (Fig.4b-c) and that myriocin increases NE deformations in human cancer cell line HCC1937 (Fig.5a).

As for the relationship between NE deformation, DSBs mobility and misrepair, when sphingolipids synthesis is altered, we have shown that shSPTLC1 increases DSBs mobility and misrepair in BRCA1-deficient MEFs treated with PARPi (Fig. 4d-g). Furthermore, shSPTLC1 or myriocin increase the sensitivity of BRCA1-deficient cells to PARPi, while they do not affect the sensitivity to PARPi of the BRCA1-proficient cells that can instead promptly and properly repair the DSBs by HR (Fig.4h-i and Extended Data Fig. 4e; Fig. 5b-f and Extended Data Fig. 5d-e). We believe these results prove that affecting the sphingolipids synthesis promotes the mobility and the mis-repair of the DSBs generated by PARPi and not repairable by HR due to the absence of BRCA1.

We would like to point out that evidence that factors affecting mobility affect also DSBs misrepair/ cell death in the context of BRCA1 deficiency, were already provided by us, for the microtubule-LINC complex (Lottersberger et al., Cell, 2015; PMID: 26544937; Figure 10 for Reviewers only)- as also recently confirmed by another study (Shokrollahi et al., NSMB, 2024; PMID: 38632359; Figure 11 for Reviewers only)- and by others, for the Schlafen protein 5 (Huang et al., Mol Cell, 2020; PMID: 36854302; Figure 12 for Reviewers only), supporting the notion that mobility has an important role in DSBs mis-repair in BRCA1-deficient cells.

[REDACTED]

Figure 10 for Reviewers only: taxol and LINC deficiency reduce the mobility of IR-induced DSBs, reduce DSBs misrepair and promotes survival in BRCA1-depleted PARP-inhibited MEFs (from Lottersberger et al., Cell, 2015)

[REDACTED]

Figure 11 for Reviewers only: depletion of LINC complex-interacting kinesines Kif5b or Kif3C increase the survival of BRCA1-deficient cancer cells to PARPi (from Shokrollahi et al., NSMB, 2024)

[REDACTED]

Figure 12 for Reviewers only: deletion of SLFN5 reduce the mobility of mCherry-53BP1-2 foci after IR in human cancer cell line U2OS and decreases the sensitivity to PARPi of MEFs depleted of BRCA1 (from Huang et al., Mol Cell, 2023)

Poit to Point Response to Reviewers

Reviewer #3 (Remarks to the Author):

The authors have addressed most of my comments, and the manuscript has significantly improved.

We thank the reviewer for the positive comment.

A few comments:

Almost every image in the manuscript is missing scale bars.

We have added the scale bars to all images.

The significance of the data in Figure 1e remains unclear. One more invagination per cell and a few EM images may surprise the readership as significant evidence to support the hypothesis postulated in the paper.

The median number of nuclear deformations in cells activating the DNA damage response due to PARPi treatment in the absence of BRCA1 has increased from one to two, representing a twofold rise, although the EM analysis allows only the visualization of a single time-point and a single Z-stack view of the nuclei, without the possibility of identifying the nuclei undergoing DNA damage response at that specific time (only 50% of the total). The ANOVA analysis confirms this increase as statistically significant, supporting our hypothesis that nuclear deformation increase after DNA damage activation.

It is unclear what the nuclear gap is. Ext fig 2c?

We have now explained in the legend what is considered nuclear gap: "Square indicates a gap (= substantial interruption in the peripheral staining pattern of Lamin-B1)"

Fluorescent microscopy of cells expressing GFP-lamin A showing the rescue of nuclear invaginations may strengthen the results in Figure 3.

As suggested, we are now showing a partial rescue of nuclear invagination after expression of GFP-hLMNA in Brca1^{F/F} Lmna^{-/-} MEFs (new Supplementary Fig.2b).